# An electricity-driven mobility circular economy with lifecycle carbon footprints for climate-adaptive carbon neutrality transformation

Aoye Song [1,2,6], Zhaohui Dan[1,6], Siqian Zheng[1,3] ✉ & Yuekuan Zhou[1,2,4,5] ✉

Under the carbon neutrality targets and sustainable development goals, emergingly increasing needs for batteries are in buildings and electric vehicles. However, embodied carbon emissions impose dialectical viewpoints on whether the electrochemical battery is environmentally friendly or not. In this research, a community with energy paradigm shifting towards decentralization, renewable and sustainability is studied, with multi-directional Vehicle-to-Everything (V2X) and lifecycle battery circular economy. Approaches are proposed to quantify the lifecycle carbon intensity of batteries. Afterwards, pathways for zero-carbon transformation are proposed to guide the economic feasibility of energy, social and governance investment behaviors. Results show that lifecycle zero-carbon battery can be achieved under energy paradigm shifting to positive, V2X interaction, battery cascade utilization and battery circular economy in various climate regions. This study proposes an approach for lifecycle battery carbon intensity quantification for sustainable pathways transition on zero-carbon batteries and carbon-neutral communities.

With the deteriorated environmental issues and accelerated climate change, countries outside of China are trying to electrify buildings and transportation systems with cleaner power production[1]. For example, they use electric vehicles to replace traditional combustion engine vehicles, use batteries to store excess renewable energy, and balance the peak and valley demand for building energy systems[2,3]. However, the battery itself is carbon-intensive, especially in the processes of manufacturing, transportation, operation and recycling[4]. For example, carbon emissions of electric vehicles (EVs) charged by traditional fossil fuel power grids will be higher than

traditional internal combustion engine vehicles (ICEVs)[5]. Therefore, during the electrification era, it is necessary to adopt various strategies to reduce the battery carbon intensity by offsetting embodied carbon via renewable energy penetration during the operation phase. Common operating strategies include peak shaving[6], power shifting[6], fast power response[7], peer-to-peer energy trading[8], vehicle-to-grid (V2G)[9], vehicle-to-building (V2B)[6], demand side management[10], and EV battery cascade utilization[11]. For example, under the carbon neutrality transition era, electrochemical batteries have been applied in various scenarios, like electric

[1]Sustainable Energy and Environment Thrust, Function Hub, The Hong Kong University of Science and Technology (Guangzhou), Nansha, Guangzhou, Guangdong, China. [2]Division of Emerging Interdisciplinary Areas, The Hong Kong University of Science and Technology, Clear Water Bay, Hong Kong SAR, China. [3]Hong Kong Productivity Council, Tat Chee Ave, Kowloon, Hong Kong SAR, China. [4]Department of Mechanical and Aerospace Engineering, The Hong Kong University of Science and Technology, Clear Water Bay, Hong Kong SAR, China. [5]HKUST Shenzhen-Hong Kong Collaborative Innovation Research Institute, Futian, Shenzhen, China. [6]These authors contributed equally: Aoye Song, Zhaohui Dan. ✉e-mail: candyz129@sina.com; yuekuanzhou@hkust-gz.edu.cn

vehicles, photovoltaic (PV)-based building prosumers, centralized/distributed storages and so on. However, there are challenges in reducing battery carbon emissions and improving its economic feasibility, including complexity in nonlinear behaviors characterization (such as battery cycling ageing[12] and dynamic performance prediction under uncertainty), transient system simulations on energy conversion and management with combined strategies, quantitative analysis on carbon reduction and environmental impact, and economic feasibility of practical applications. Therefore, how to achieve carbon neutrality of batteries in a cost-effective manner is worth exploring.

Considering the distributed solar power supply from buildings and controllable grid power for battery charging, the energy interaction between electric vehicles and buildings or grids is an important way to efficiently reduce carbon emissions with high renewable penetration and prolong battery lifetime for sustainability[13]. Existing studies have found that as buildings gradually transform from energy consumers to prosumers, an interactive building-vehicle energy network can address the intermittency of renewable energy with high renewable energy self-consumption[6]. Secondly, battery cascade utilization is a cost-effective method to reduce battery carbon emissions, because EV battery reuse in other scenarios (e.g., centralized PV farms, buildings, etc.) can prolong battery service lifetime, improve battery utilization, and store more renewable energy[14,15]. Moreover, secondary reuse batteries can store off-peak electricity from the grid to solve the mismatch between supply and demand in building energy systems. Therefore, it is worth further exploring the potential for achieving carbon neutrality of batteries based on the battery circular economy in different climate conditions. Additionally, quantifying the contribution of advanced vehicle-to-everything (V2X) interactions and the cascading utilization of batteries to the carbon neutrality industry is also worth investigating.

In this work, a multi-energy district community with an energy paradigm shifting towards decentralization, renewable and sustainability is studied, with multi-directional V2X interaction, battery cycling ageing, lifecycle battery circular economy, economics and carbon emissions throughout the entire battery lifetime. Then, four corresponding models on the battery circular economy are developed, i.e., a battery degradation model, a transient building-vehicle energy network model, a cascade EV battery utilization model, and a calculation model on battery carbon intensity. Afterwards, pathways to battery carbon neutrality are explored under different climate types. Results show that the replacement of internal combustion engine vehicles (ICE-cars and ICE-buses) with electric vehicles (E-cars and E-buses) cannot effectively reduce lifecycle carbon emissions unless the operational carbon can be partially offset by cleaner energy with clean power grid upgrade or renewable energy integration. Furthermore, compared with the traditional district energy community scenario (relying on traditional fossil fuel power plants), the formulated district community with interactive energy sharing and EV battery cascade utilization under the battery circular economy can significantly reduce carbon emissions and achieve better economic benefits. Synergistic adoption of interactive energy sharing and EV battery cascade utilization can significantly reduce the battery carbon intensity towards negative carbon emission and substantially improve economic performances. A more detailed methodological discussion is shown in Supplementary Fig. 1 and Supplementary Note 1. Lastly, lifecycle zero-carbon batteries can be achieved with embodied carbon (i.e., raw materials, manufacture and assembling processes) completely offset by operational carbon in most climate types by 1) distributed building-integrated photovoltaics (BIPVs) and wind turbines in building prosumers; 2) managing multi-energy network with V2X interactions; 3) cascade EV battery utilization for service lifetime extension; 4) power grid upgrade from traditional coals to cleaner power grids.

## Results

### Battery lifecycle and circular economy
Generally, traditional battery lifecycle process is divided into five phases[4]: raw materials mining, manufacturing, operation, reuse and recycling phase, as shown in Fig. 1a. During the raw material mining and manufacturing phase, batteries are manufactured and then used in electric vehicles (EVs). During the operation phase with advanced V2X interactions, as shown in Fig. 1c, EV batteries can not only provide energy for the traveling of EVs but also help to store renewable energy and achieve energy sharing between buildings as mobile energy storages until their relative capacity drops to 80%. The principle of multi-directional V2X interactions can be explained by Supplementary Figs. 2–5 and Supplementary Note 2. When renewable energy is higher than the demand, the excess renewable energy is stored in battery energy storage systems (including electric vehicles). When renewable energy is lower than the demand, static batteries and electric vehicles will supply energy to cover the demand. During the battery reuse phase as shown in Fig. 1d, the cascade EV battery utilization technology can reuse EV batteries with a relative capacity of 80%. After recycling the damaged battery cells, the remaining parts can be used as reused batteries for energy storage in renewable energy power stations, peak load shifting and valley filling in buildings, etc., until their relative capacity drops to 60%. Finally, in the battery recycling phase, the batteries with a relative capacity of less than 60% will be recycled[16].

Figure 1b shows the carbon intensity in the battery circular economy. The embodied carbon emission in raw materials, in the battery manufacturing process, and generated during battery recycling are 51, 34, and 69.8 kg $CO_{2,e}$ $kWh^{-1}$ ($CO_{2,e}$ is the abbreviation for carbon dioxide equivalent), respectively[17]. Note that, due to the complexity and inconsistency of data sources, the minimum values were chosen to maximize the possibility of achieving the "net-zero carbon" result. Furthermore, as carbon emissions in the transportation stage only account for 1% of the entire lifecycle of Li-ion batteries[17] and uncertainties in specific transportation distances, transportation of the renewable and battery systems was not included in the lifecycle calculations. However, the carbon intensity quantification is rather difficult in both the operation and reuse phases. This is because that these two phases involve different charging sources (fossil power and renewable energy) under flexible operational modes (vehicle-to-grid, vehicle-to-building, etc.), while the different battery ageing magnitudes will also lead to the difficulty in carbon emissions calculation.

### Lifecycle carbon emission analysis for vehicles
The current era focuses on building and transportation transformation towards electrification, and ICEVs are gradually replaced by EVs, together with the mainstream for the rapid increase in electrified vehicles. However, before widely promoting the E-mobility in transportation, an urgent question needs to be discussed and declared whether EVs are really cleaner than ICEVs or not. Otherwise, the blind development with misleading priority on E-transportation will conversely increase carbon emissions, with associated battery waste, environmental pollution and disposal issues.

Figure 2a shows the comparing between the carbon emissions of EVs and ICEVs (i.e., private cars and shuttle buses) during the operation phase and the lifecycle process. It is noted that, during the operation phase, the carbon emissions of electric cars (E-cars) (112.0–131.6 g $CO_{2,e}$ $km^{-1}$) are much lower than those of internal combustion engine cars (ICE-cars) (169.5 g $CO_{2,e}$ $km^{-1}$). However, when considering the lifecycle carbon emissions, the difference in lifecycle carbon emissions of E-cars (193.6–213.2 g $CO_{2,e}$ $km^{-1}$) and ICE-cars (213.3 g $CO_{2,e}$ $km^{-1}$) is not so obvious, since the carbon emissions of EVs (77.14 g $km^{-1}$) during the manufacturing and recycling stages are higher than those of ICEVs (43.80 g $km^{-1}$)[18]. For shuttle buses, as shown in Fig. 2a, lifecycle of internal combustion engine buses (ICE-buses) at 721.3 g $CO_{2,e}$ $km^{-1}$ is lower than those of electric buses (E-buses) at 715.7–784.4 g $CO_{2,e}$ $km^{-1}$,

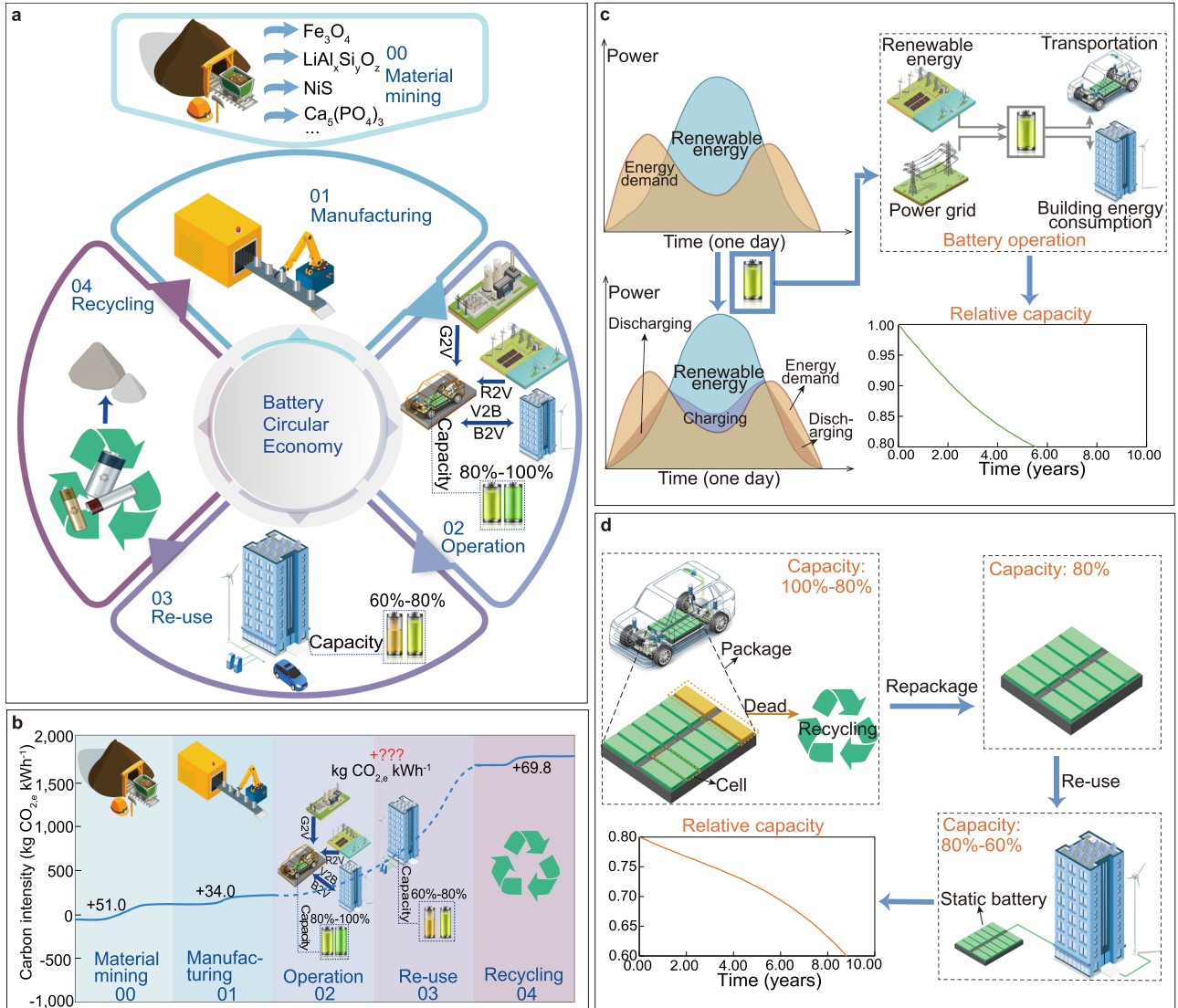

**Fig. 1 | Schematic diagram of battery circular economy with cascade battery utilization. a** Battery circular economy with five different phases (materials mining, manufacturing, operation, reuse and recycling); **b** Carbon intensities at different phases of battery lifecycle; **c** Schematic diagram of battery operational phase and its degradation curve; **d** Schematic diagram of battery reuse phase and its degradation curve. Vehicle-to-everything (V2X) interactions are shown in the operational phase as grid-to-vehicle (G2V), renewable-to-vehicle (R2V), building-to-vehicle (B2V) and vehicle-to-building (V2B). The carbon emissions during the operational phase are displayed as "+???" due to difficulties in quantification.

together with the carbon emissions during the operation phase at 391.2–459.9 g $CO_{2,e}$ km⁻¹ for E-buses and 436.6 g $CO_{2,e}$ km⁻¹ for ICE-buses.

The transition from traditional power grids to clean power grids can significantly reduce the operational carbon and lifecycle carbon emissions of E-cars. For example, the carbon emission factor (CEF) in Guangzhou has decreased from 0.8652 to 0.4263 as shown in Fig. 2b, and the lifecycle carbon emissions of E-cars have dropped rapidly from 193.6–213.2 g $CO_{2,e}$ km⁻¹ (Fig. 2a) to 101.6–194.6 g $CO_{2,e}$ km⁻¹ (Fig. 2c). In addition, the lifecycle carbon emissions of E-buses have dropped rapidly from 715.7–784.4 g $CO_{2,e}$ km⁻¹ (Fig. 2a) to 394.6–719.3 g $CO_{2,e}$ km⁻¹ (Fig. 2c). This is mainly due to the continuous promotion of renewable energy in local power grids for cleaner power grid upgrade in different climate regions. According to the research by Li et al.[18] energy structures of a future clean grid in different climate zones are shown in Fig. 2d, in which Guangzhou now has 30.20% hydropower and 4.28% wind, photovoltaic, and biomass power generation.

## Carbon emission factor on various renewable systems

Renewable systems supply clean energy without carbon emission during the operational stage, while embodied carbons on raw materials, manufacture and recycling processes need to be considered[19]. As part of measuring the battery carbon intensity throughout its their entire lifecycle process, the embodied carbon emissions of renewable systems cannot be ignored, and it is defined as the amount of carbon emitted during construction, operation, and recycling processes.

Figure 3 shows the abundance of wind (Fig. 3a, b) and solar energy (Fig. 3c) resources in China for different provinces located in five climate regions. Both wind and solar resources in China are relatively abundant in the west and north, and insufficient in the developed eastern regions. Among the large cities in this study, Shanghai has the most abundant wind resources, with an average wind speed of 5.63 m s⁻¹ and an average wind power density of 190.41 W m⁻². Beijing is relatively rich in solar energy resources, with an annual horizontal radiation of about 1527.60 kWh m⁻².

The embodied carbon emission factors (ECEFs) for solar PV, BIPVs and wind turbines (WTs) across different climates are calculated as

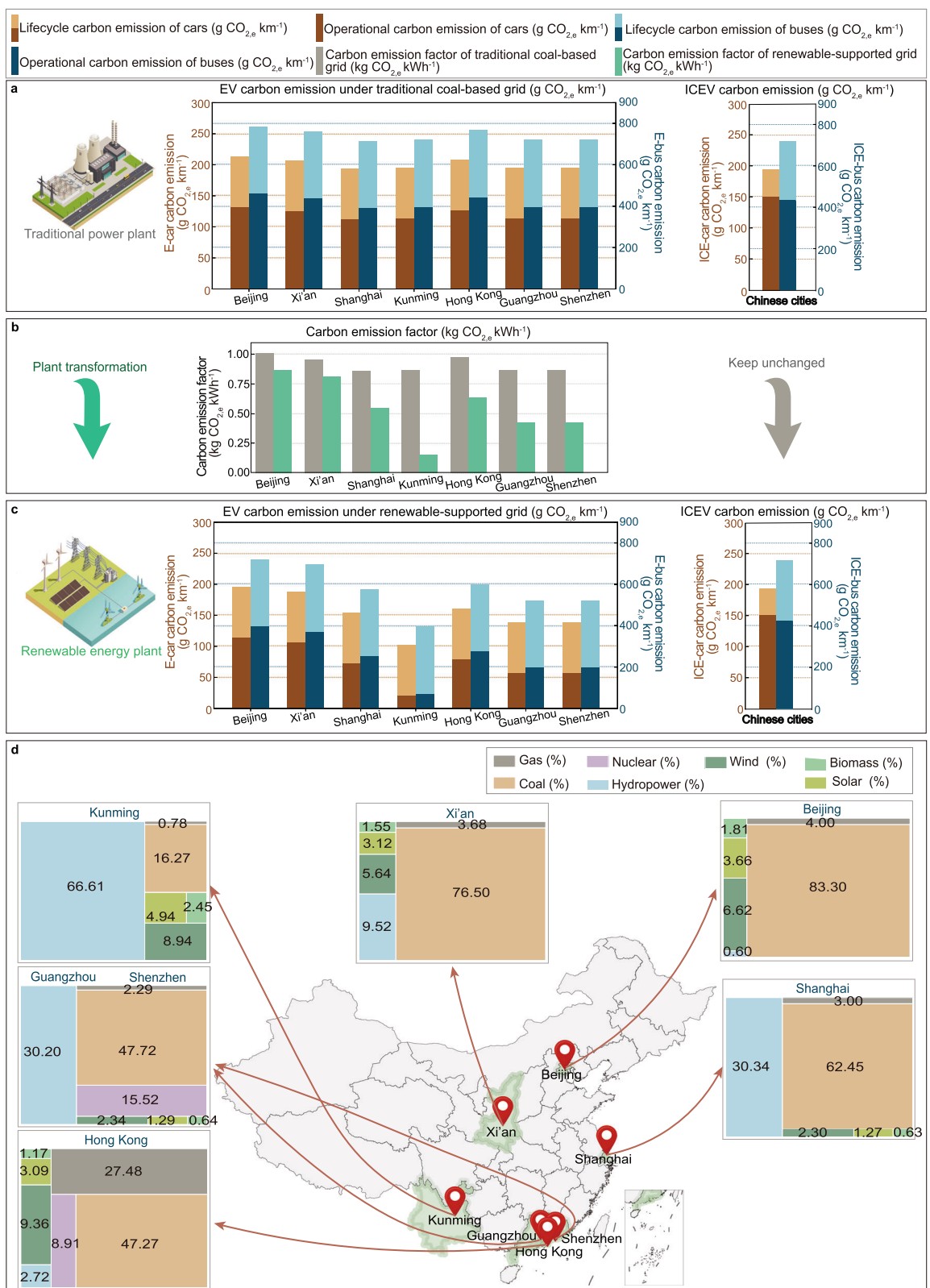

**Fig. 2 | Comparison of lifecycle carbon emissions of electric vehicles (EVs) and internal combustion engine vehicles (ICEVs) under different grid energy compositions. a** Carbon emission per kilometer of EVs and ICEVs in operation phase and lifecycle process under different climate zones; **b** Changes in carbon emission factor from the traditional grid to a clean grid; **c** Changes in the lifecycle carbon emissions of EVs and ICEVs with the transformation from a traditional grid to a clean grid; **d** Energy structures of the current clean grid in different climate zones.

shown in Fig. 3d. The lowest ECEF$_{PV}$ and ECEF$_{BIPV}$ are in Kunming at 0.0205 kg CO$_{2,e}$ kWh$^{-1}$ and 0.0570 kg CO$_{2,e}$ kWh$^{-1}$, respectively. The highest ECEF$_{PV}$ and ECEF$_{BIPV}$ are in Shenzhen at 0.0281 kg CO$_{2,e}$ kWh$^{-1}$ and 0.0669 kg CO$_{2,e}$ kWh$^{-1}$, respectively. For ECEFs of wind turbines, in areas with high annual average wind speeds, such as Hong Kong and Shenzhen, the ECEF$_{WT}$ is between 0.0118 kg CO$_{2,e}$ kWh$^{-1}$ and 0.0286 kg CO$_{2,e}$ kWh$^{-1}$. However, in areas with low annual mean wind speeds, such as Kunming, the ECEFs of wind turbines is high at 0.153 kg CO$_{2,e}$ kWh$^{-1}$.

## Battery circular economy in district communities

Depending on various application scenarios, electrochemical battery plays significant roles, such as daily transportation in electric vehicles, renewable energy self-consumption in the PV-battery scenario[20], grid power stabilization and frequency regulation in the PV-battery-grid scenario[21], demand response and energy sharing in building-EV-building scenarios[22]. To explore pathways for zero-carbon battery transformations, five typical scenarios are studied, as shown in Fig. 4. Traditional scenario (Scenario A, Fig. 4a) assumes that EVs are charged by the grid with power transmitted from coal-based power plants, and there is no interaction between buildings and EVs. This scenario will be used as a reference scenario. A pure renewable supported scenario (Scenario B, Fig. 4b) assumes that EVs are fully supported by solar energy and there is no interaction between buildings and EVs. The multi-directional V2X scenario (Scenario C, Fig. 4c) assumes that there are energy interactions between multiple buildings and EVs, and EVs can spatially share energy among geographically different buildings. The Cascade EV battery reuse scenario (Scenario D, Fig. 4d) assumes that EV batteries are repurposed as energy storage batteries for buildings after their relative capacity has dropped to 80% of their initial capacity. Synergistic scenario (Scenario E, Fig. 4e) combines the multi-direction V2X scenario with the cascade EV battery reuse scenario for energy storage in buildings, when the EV battery retires with its relative capacity dropped to 80% of the initial capacity. Note that in all scenarios, the zero-energy paradigm is studied, i.e., annual total renewable generation from BIPVs and on-site wind turbines equal to the annual electricity consumption of buildings and EVs.

In this section, both net present value (NPV) and battery carbon intensity are analyzed to illustrate the economic feasibility and zero-carbon pathway, respectively. Note that the NPVs of other scenarios are calculated based on the traditional scenario.

In Scenario A (Traditional Scenario), the battery carbon intensity is positive in all climate conditions (Fig. 5a–g), with the lowest value appearing in Shanghai (Fig. 5e) at 1604.62 kg CO$_{2,e}$ kWh$^{-1}$ and the highest value in Beijing at 1857.08 kg CO$_{2,e}$ kWh$^{-1}$. This value is highly correlated with the coal-based grid CEF[18] as shown in Fig. 2b. In the traditional scenario, the energy for charging electric vehicles comes entirely from the coal-based power plant. The higher grid carbon emission factor will directly lead to a higher battery carbon intensity, especially during the operational stage.

In Scenario B (Pure renewable supported scenario), the battery carbon intensity is negative in all climate conditions (Fig. 5a–g). This is because of the replacement of coal-based power by cleaner power, and the carbon reduction effect of renewable energy is more obvious in regions with a higher coal-based grid emission factor. With respect to the NPV, the results vary from 13.59 × 10$^3$ to 83.94 × 10$^3$ US$ (Fig. 5a–g), which is because electricity prices are generally higher than the cost of solar power generation in different climate regions. The results of Scenario B illustrate that both a significant reduction in the battery carbon intensity and an improvement in economic benefits can be achieved with 100% renewable energy in the cleaner power grid. However, this pure renewable supported scenario is difficult to achieve in reality, due to the currently low proportion of renewable systems and intermittence of renewable energy. Therefore, this scenario is set

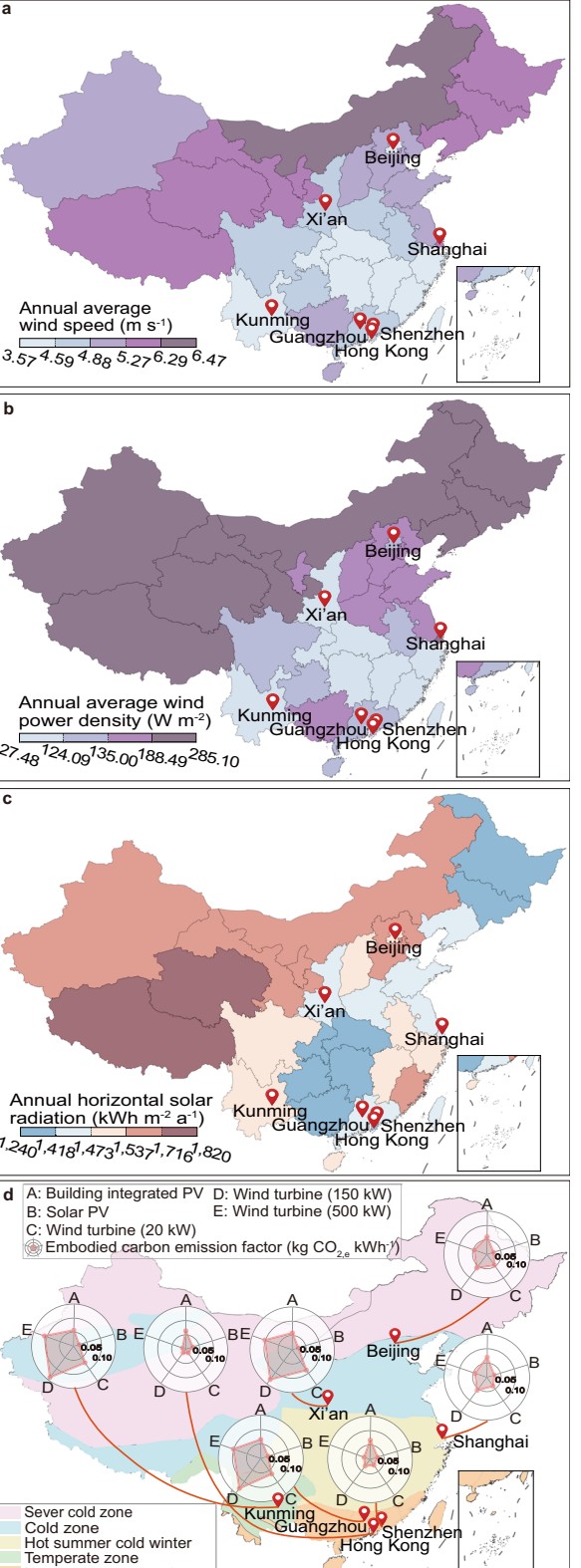

**Fig. 3 | Distribution and embodied carbon emission factors (ECEFs) of renewable energy in different regions of China. a** The annual average wind speed (m s$^{-1}$) of each province in China at 70 m height; **b** The annual average wind power density (W m$^{-2}$) of each province in China at 70 m height; **c** The annual horizontal solar radiation (kWh m$^{-2}$ a$^{-1}$) of each province in China; **d** ECEFs (kg CO$_{2,e}$ kWh$^{-1}$) of photovoltaics (PVs) and wind turbines under various climate regions.

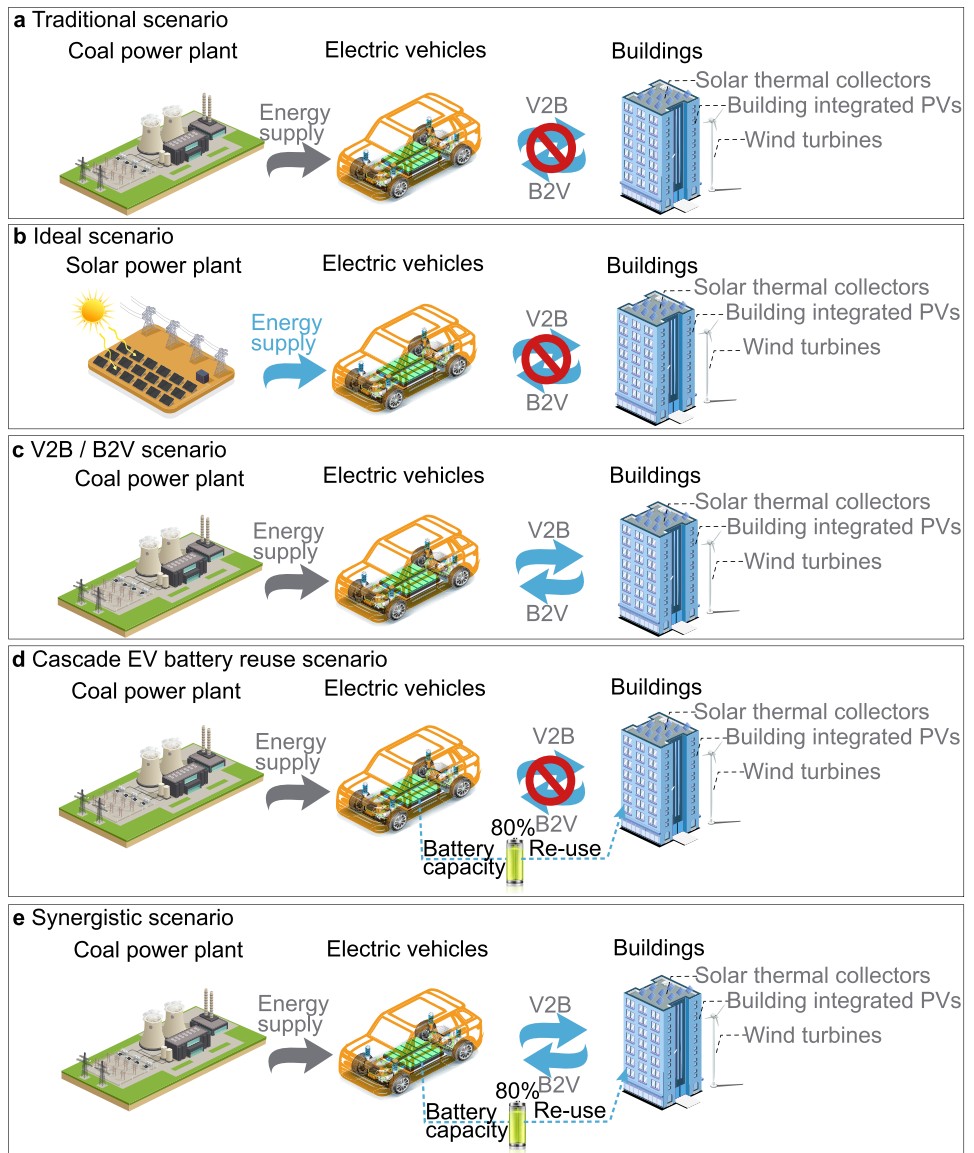

**Fig. 4 | Different scenarios of building-transportation shared energy network.**
**a** Traditional scenario (Scenario A) powered by coal power plants without vehicle-to-everything (V2X) interaction; **b** Pure renewable supported scenario (Scenario B) without V2X interaction; **c** Multi-direction vehicle-to-everything (V2X) scenario (Scenario C) with building-to-vehicle (B2V) and vehicle-to-building (V2B) interaction; **d** Cascade electric vehicle (EV) battery reuse scenario (Scenario D); **e** Synergistic scenario (Scenario E) with V2X interaction and Cascade EV battery reuse. Buildings in all scenarios are equipped with solar thermal collectors, on-site wind turbines and building-integrated photovoltaic (PV) panels.

as an ideal scenario with the minimum threshold in battery carbon intensity.

In Scenario C (Multi-direction V2X scenario), the battery carbon intensity is greatly reduced compared to Scenario A. For example, the battery carbon intensity is reduced from 1857.08 to −134.47 kg $CO_{2,e}$ $kWh^{-1}$ in Beijing (Fig. 5c). This is mainly because that, the power interaction of EVs with building energy can enable a large amount of renewable energy from BIPV and on-site wind turbines to be charged in EVs ($1.36 \times 10^6$ kWh in Beijing), so as to reduce the power import from the grid (i.e., from $8.18 \times 10^7$ kWh to $8.05 \times 10^7$ kWh in Beijing). Meanwhile, NPVs show positive value in Scenario C in most regions. This indicates that the multi-direction V2X interaction can improve the economic feasibility of the integrated system, with the overwhelming dominance of import cost savings (i.e., the reduced amount of electricity imported from the power grid multiplied by the electricity price) over the battery cycling ageing costs. Note that, compared to

other cities, the NPV in Kunming (Fig. 5a) is relatively lower. The main reason for the lowest NPV for Kunming in Scenario C is that EVs store more renewable energy in Kunming, resulting in higher battery degradation costs compared to other regions (i.e., $1.28 \times 10^5$ US$ in Kunming, and $0.88-0.90 \times 10^5$ US$ in other regions).

Considering the remaining storage capacity of retired EV batteries, cascade EV battery reuse for renewable energy storage in the buildings was proposed and studied from the lifecycle perspective. This strategy involves using repurposed EV batteries as energy storage batteries for zero-energy buildings and energy storage power stations after their capacity drops to 80%. Results shown in Fig. 5a−g show that compared with Scenario A, the EV battery cascade utilization strategy can effectively reduce the battery carbon intensity and slightly increase the NPV. The reason is that the excess renewable energy generated by BIPVs and on-site wind turbines can be stored by cascade EV batteries reused in buildings, thereby reducing battery

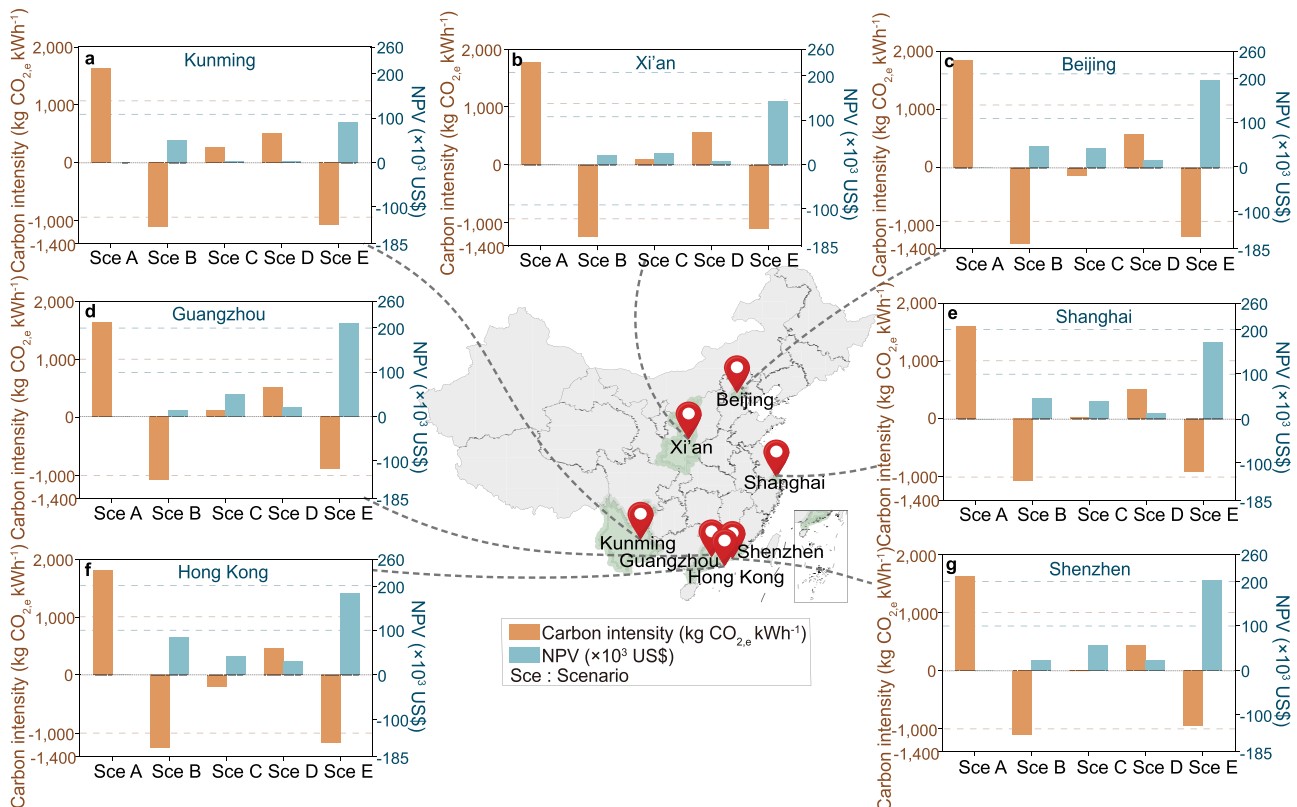

**Fig. 5 | Lifecycle battery carbon intensity and net present values (NPVs) in different climate regions. a** Lifecycle battery carbon intensity and NPVs in Kunming; **b** Lifecycle battery carbon intensity and NPVs in Xi'an; **c** Lifecycle battery carbon intensity and NPVs in Beijing; **d** Lifecycle battery carbon intensity and NPVs in Guangzhou; **e** Lifecycle battery carbon intensity and NPVs in Shanghai; **f** Lifecycle battery carbon intensity and NPVs in Hong Kong; **g** Lifecycle battery carbon intensity and NPVs in Shenzhen. Results for each city include Scenario A (Sce A), Scenario B (Sce B), Scenario C (Sce C), Scenario D (Sce D) and Scenario E (Sce E).

carbon intensity and leading to import cost savings. However, compared to Scenario C, Scenario D (cascade EV battery reuse scenario) is less superior in battery carbon intensity, as indicated by the much higher battery carbon intensity in Scenario D than that in Scenario C. In other words, due to the lower cost of reused batteries (35 US$ kWh⁻¹)[23] than new batteries (125 US$ kWh⁻¹)[24], the cascade EV battery reuse technology can effectively improve the NPV with economic feasibility through the extension in battery service lifetime, while multi-direction V2X interaction exerts high impact on the battery carbon intensity.

Based on the above analysis results, this section aims to combine the advantages of both economic feasibility from cascade EV battery reuse technology and battery carbon intensity from multi-direction V2X interaction, forming a synergistic scenario for techno-economic-environmental competitiveness. As shown in Fig. 5a–e, compared to Scenario A, the proposed multi-direction V2X energy sharing with cascade EV battery reuse (Scenario E) significantly reduces the carbon intensity to negative values, demonstrating the effectiveness for zero- or even negative- battery carbon intensity pathway transitions. Meanwhile, the NPV can be improved from 0 (Scenario A) to 90.73–144.03 × 10³ US$ (Scenario E). This demonstrates the economic feasibility of zero- or even negative- battery carbon intensity pathway transitions. Furthermore, in Scenario E, the battery carbon intensity closely approaches the level of Scenario B (the pure renewable supported scenario), and the NPV even exceeds that of Scenario B. This indicates that battery carbon neutrality with economic competitiveness can be effectively achieved through the combination of multi-direction V2X with the cascade reuse strategy on retired EV batteries. In addition, Supplementary Table 1, Supplementary Figs. 6–8 and

Supplementary Note 3 describes the impact of battery charge and discharge cycles and depth of discharge on the degradation and carbon intensity under different scenarios.

In addition, the combination of multi-direction V2X with the cascade EV battery reuse strategy can not only greatly reduce the carbon intensity in the operation phase, but also reduce the carbon intensity in the manufacturing and recycling phases. Figure 6 shows changes in carbon intensity and battery degradation curve before and after adopting the multi-direction V2X interaction with the cascade EV battery reuse strategy in Guangzhou. In the absence of battery reuse and V2X interaction, as shown in Fig. 6a, the battery degradation is slowe and it is recycled when the relative capacity drops to 80%. Through V2X interaction and battery reuse, as shown in Fig. 6b, the battery degradation accelerates, but it can be utilized until 60% before being recycled. Additionally, it is noted that in the absence of battery reuse and V2X interaction, Fig. 6c exhibits high carbon intensity during different phases. However, in Fig. 6d, the battery's carbon intensity decreases rapidly after V2X interaction and battery reuse. The main reason for the decrease in the raw material and manufacturing phase is that the embodied carbon emission of batteries with 80%–60% RC will be excluded after 20 years of simulation. As for the decrease of the carbon intensity in the recycling phase, the reason is that with the adoption of the cascade EV battery reuse strategy, the battery recycling progress is delayed and the proportion of recycled batteries to the total number of batteries is reduced during the simulated 20 years. Moreover, the carbon intensity in the operation phase has dropped significantly, which shows the effectiveness of the multi-direction V2X interaction with the cascade EV battery reuse strategy in the carbon neutrality transformation.

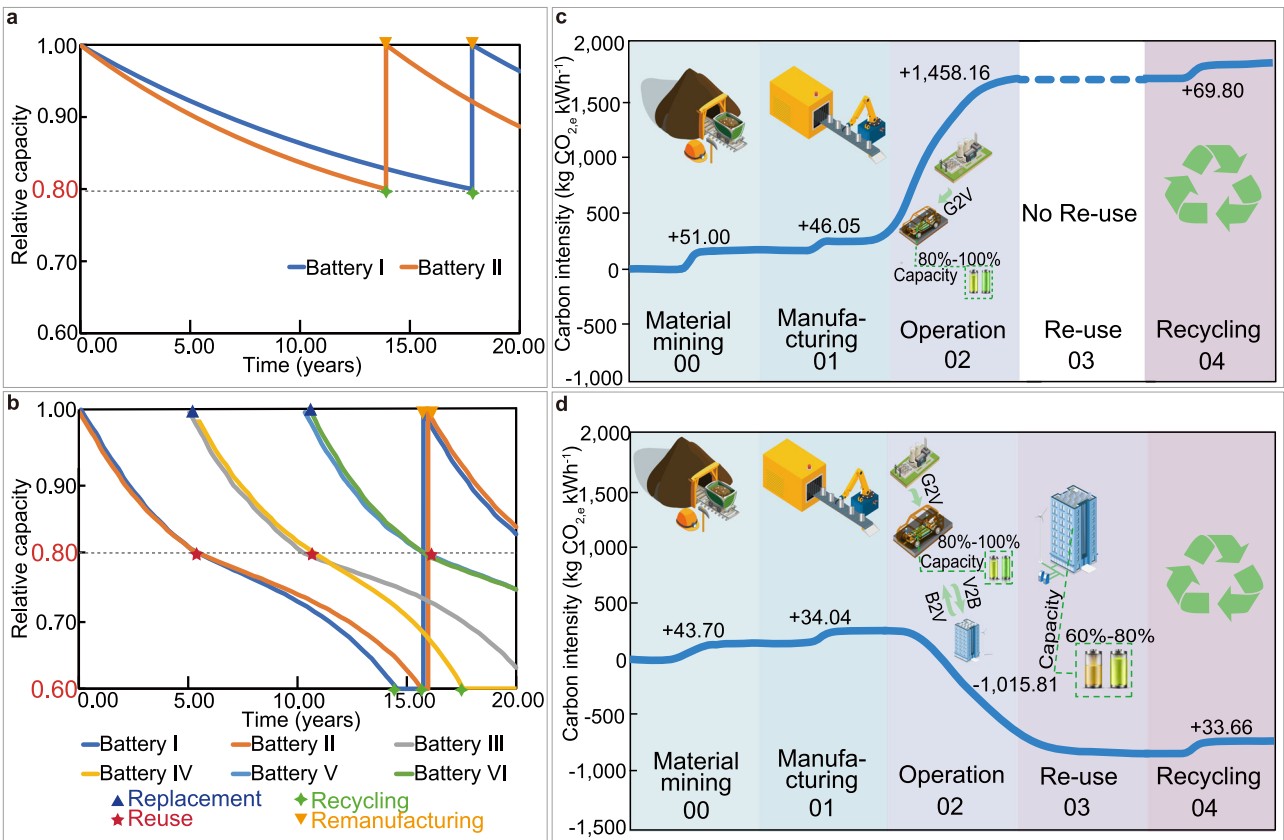

**Fig. 6 | Evolution of battery degradation and battery carbon intensity in various phases in Guangzhou. a** The degradation curve in Scenario A; **b** The degradation curve in Scenario E; **c** The battery carbon intensity of different phases in Scenario A; **d** The battery carbon intensity of different phases in Scenario E. Vehicle-to-everything (V2X) interactions are shown in the operational phase as grid-to-vehicle (G2V), building-to-vehicle (B2V) and vehicle-to-building (V2B).

## Battery carbon neutrality on different energy paradigms

Considering the high correlation among NPV, battery carbon intensity, renewable energy generation and total system demand, the impact of different energy paradigms on NPV and battery carbon intensity has been studied and analyzed. In order to explore and calculate the degree of this effect, three kinds of paradigms are set according to the relative difference between renewable generation and total energy demand, i.e., net-zero energy paradigm (Paradigm I), low positive-energy paradigm (Paradigm II) and high positive-energy paradigm (Paradigm III), as shown in Fig. 7e. In the Paradigm I, the system consumes as much energy as it produces. In Paradigms II and III, the system produces 1.5 and 2 times as much energy as it consumes, respectively. In addition, depending on the differences in energy consumption and power generation among different buildings, different types of prosumers are defined, i.e., positive prosumers (buildings with more power generation than energy consumption), net-zero prosumers (buildings with power generation close to energy consumption) are referred as, and negative prosumers (buildings with power generation less than energy consumption), as shown in Fig. 7a–d.

The battery carbon intensity and NPV for each paradigm in different climate conditions are demonstrated in Fig. 7f. Generally, the transition from net-zero to positive energy paradigm will reduce both battery carbon intensity and NPV. For example, in Beijing, the highest NPV values appear in the net-zero energy paradigm at $0.20 \times 10^6$ US\$ with a battery carbon intensity at $-1192.12$ kg $CO_{2,e}$ kWh$^{-1}$. In addition, compared to the slight decrease in NPV in Shenzhen, NPV decreases more obviously along with the decrease in battery carbon intensity in Beijing, Guangzhou, Kunming and Xi'an regions. This is because wind

resources are scarce in inland regions, leading to the installation and maintenance costs of wind turbines overweighing the economic benefits of renewable energy supply. However, a completely opposite tendency can be noticed in Hong Kong. Results show that in Hong Kong, the NPV is highest at Paradigm II, and the carbon intensity is $-1206.42$ kg $CO_{2,e}$ kWh$^{-1}$ with the highest NPV at $0.43 \times 10^6$ US\$. This is because Hong Kong has abundant renewable resources (especially wind resources), which balance the costs of renewable energy equipment (i.e., the operating cost savings is $1.79 \times 10^6$ US\$, which is much higher than the installation and maintenance costs of wind turbines at $1.17 \times 10^6$ US\$.). Overall, lifecycle battery carbon neutrality can be achieved through multi-directional V2X interaction and EV battery cascade utilization under the lifecycle battery circular economy within a net zero energy community. Furthermore, in areas with rich renewable resources (such as coastal cities, like Hong Kong and Shenzhen), the renewable energy harvesting systems can achieve a lower battery carbon intensity and a higher net present value, with operating cost savings overwhelming the installation and maintenance costs.

## An overall perspective of the wider region

The multiple methods proposed in this article for battery carbon neutrality transformation were further verified and promoted from a wider perspective. New York (Fig. 8a), Berlin (Fig. 8b) and Singapore (Fig. 8c) from different continents are selected. Under the traditional scenario (Scenario A), the emissions of the three cities are relatively high (1925.17, 1628.05, and 1703.89 kg $CO_{2,e}$ kWh$^{-1}$ for New York, Berlin and Singapore, respectively). In the ideal scenario (Scenario B), all cities can achieve negative carbon intensity. Furthermore, with the implementation of strategies (such as multi-direction V2X interaction

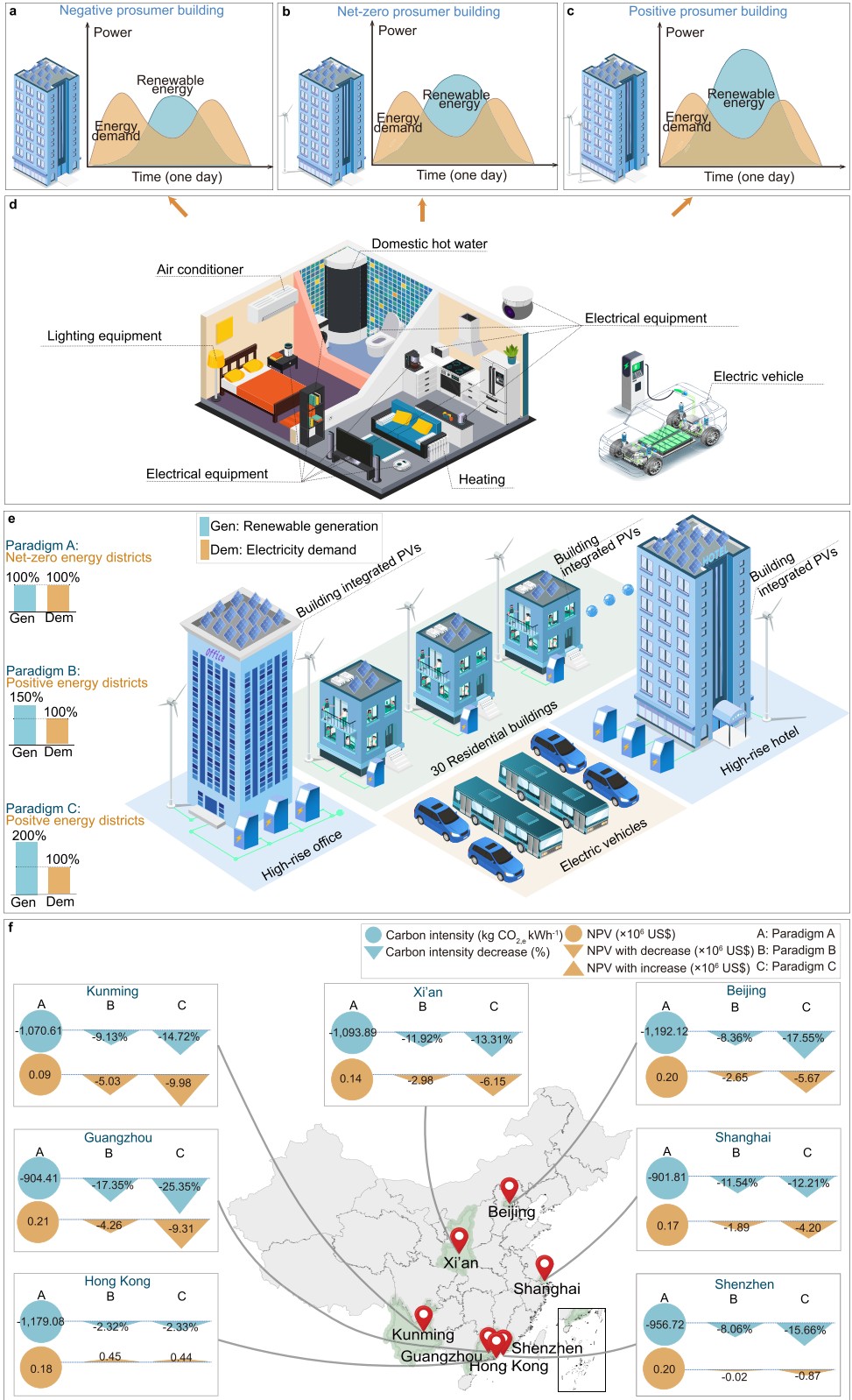

**Fig. 7 | Building-vehicle energy sharing network and its economic and environmental analysis under different energy paradigms. a** Negative prosumer building; **b** Net-zero prosumer building; **c** Positive prosumer building; **d** Schematic diagram of building equipment and energy consumption; **e** Three energy paradigms of district energy communities; **f** Net present values (NPVs) and battery carbon intensity for different energy paradigms.

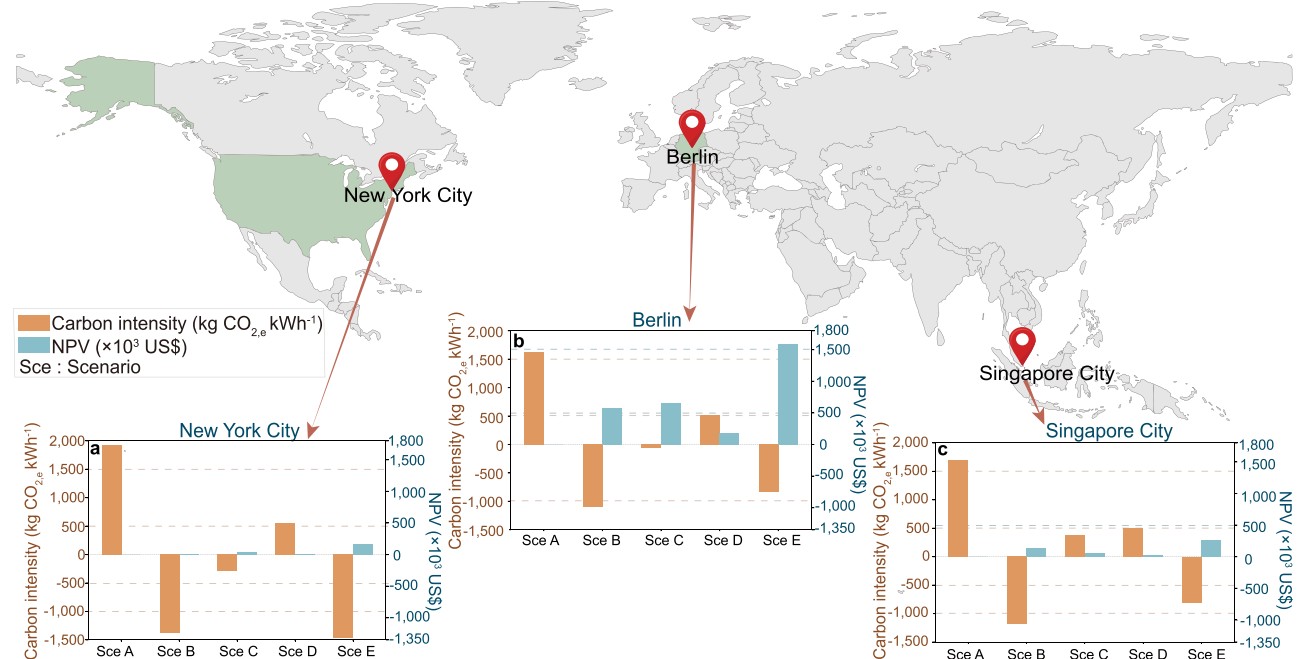

**Fig. 8 | Lifecycle battery carbon intensity and net present values (NPVs) in regions outside of China. a** Lifecycle battery carbon intensity and NPVs in New York; **b** Lifecycle battery carbon intensity and NPVs in Berlin; **c** Lifecycle battery carbon intensity and NPVs in Singapore. Results for each city include Scenario A (Sce A), Scenario B (Sce B), Scenario C (Sce C), Scenario D (Sce D) and Scenario E (Sce E).

and battery cascade utilization), carbon intensity can be significantly reduced compared to the Scenario A, as shown in Scenario C and Scenario D. The Scenario E with simultaneous adoptions of two strategies (multi-direction V2X interaction and battery cascade utilization) achieved carbon intensity close to the ideal scenario.

In terms of NPV, based on the results of Scenarios C, D and E, it can be concluded that both V2X interaction and battery cascade utilization are profitable. However, it is noted that the NPV of Berlin is much higher than that of other cities. This is because the electricity prices in Berlin are much higher (0.700 US\$ kWh⁻¹)[25] than those in New York (average at 0.216 US\$ kWh⁻¹)[26] or Singapore (average at 0.200 US\$ kWh⁻¹)[27], which makes the import cost saving in Scenarios B, C, D and E very high, resulting in a high NPV.

In general, the zero-carbon path and its analysis method represented by multi-direction V2X interaction, battery cascade utilization and lifecycle circular economy can not only guide the energy transformation of various regions in China but also provide suggestions to energy and environmental policymakers in regions outside of China.

**Impact of tax reduction exemption incentives**
Production tax credit (PTC) and investment tax credit (ITC) are two policies announced by the U.S. government to promote clean energy. Among them, PTC stated that taxes can be reduced when clean energy generates electricity, and the amount is 0.015 US\$ kWh⁻¹ (only in the first 10 years)[28] as shown in Fig. 9b. ITC stated that investors can have reduced taxes when investing in PV panels, and the reduced amount is 30% currently[28]. These two policies are mutually exclusive, and investors can only choose one of them.

There are similar policies in China. For example, the Chinese government will exempt photovoltaic and wind power products from 50% of the value-added tax (the rate is 13%) as shown in Fig. 9c. This policy is similar to the ITC. However, China currently does not implement any policy similar to PTC. Therefore, this subsection will also explore the feasibility of implementing similar policies in China based on the U.S. PTC policy, and provide guidance and suggestions for future policy development.

Based on the above assumptions, the benefits that can be brought by implementing similar ITC and PTC policies across China, as shown in Fig. 9a. In Scenario E, the NPV of various places was -90.73−211.56 × 10³ US\$. After the introduction of the ITC policy, the NPV increased to 287.22−489.89 × 10³ US\$. After the PTC policy was used, the NPV increased to 389.97−835.75 × 10³ US\$. It should be noted that due to the high tax exemption limit of the PTC policy, PVs and wind turbines are equivalent to zero tax after the introduction of the PTC policy.

The above results show that using tax reduction or exemption policies can significantly increase investment returns and enhance investors' confidence. However, how to better design PTC-like policies in China requires further discussion. This study can provide guidance and suggestions for future policy development in China or other regions.

## Discussion
In this work, a multi-energy district community with energy paradigm transformations towards decentralization, renewable and sustainability has been studied, with multi-directional V2X interaction and lifecycle battery circular economy. A general approach is proposed to quantify the lifecycle carbon intensity of batteries with V2X interactions, dynamic cycling ageing and battery cascade utilization. Battery circular economy within renewable energy-sharing communities was proposed and formulated, including vehicle-to-building (V2B) discharging, building-to-vehicle (B2V) charging, EV battery reuse, PV-battery storage and retired battery recycling. The proposed approach is then generalized and scalable in five climate regions with different solar-wind energy resources in China. The net present value with battery replacement frequencies, battery cycling ageing and electricity policies is calculated to guide the economic feasibility of energy, social and governance (ESG) investment behaviors. Pathways for lifecycle zero-carbon battery transformation have been explored and provided, together with associated net present values for economic feasibility analysis. Conservative analysis was conducted with the minimum values to maximize the possibility of achieving the "net zero carbon" result.

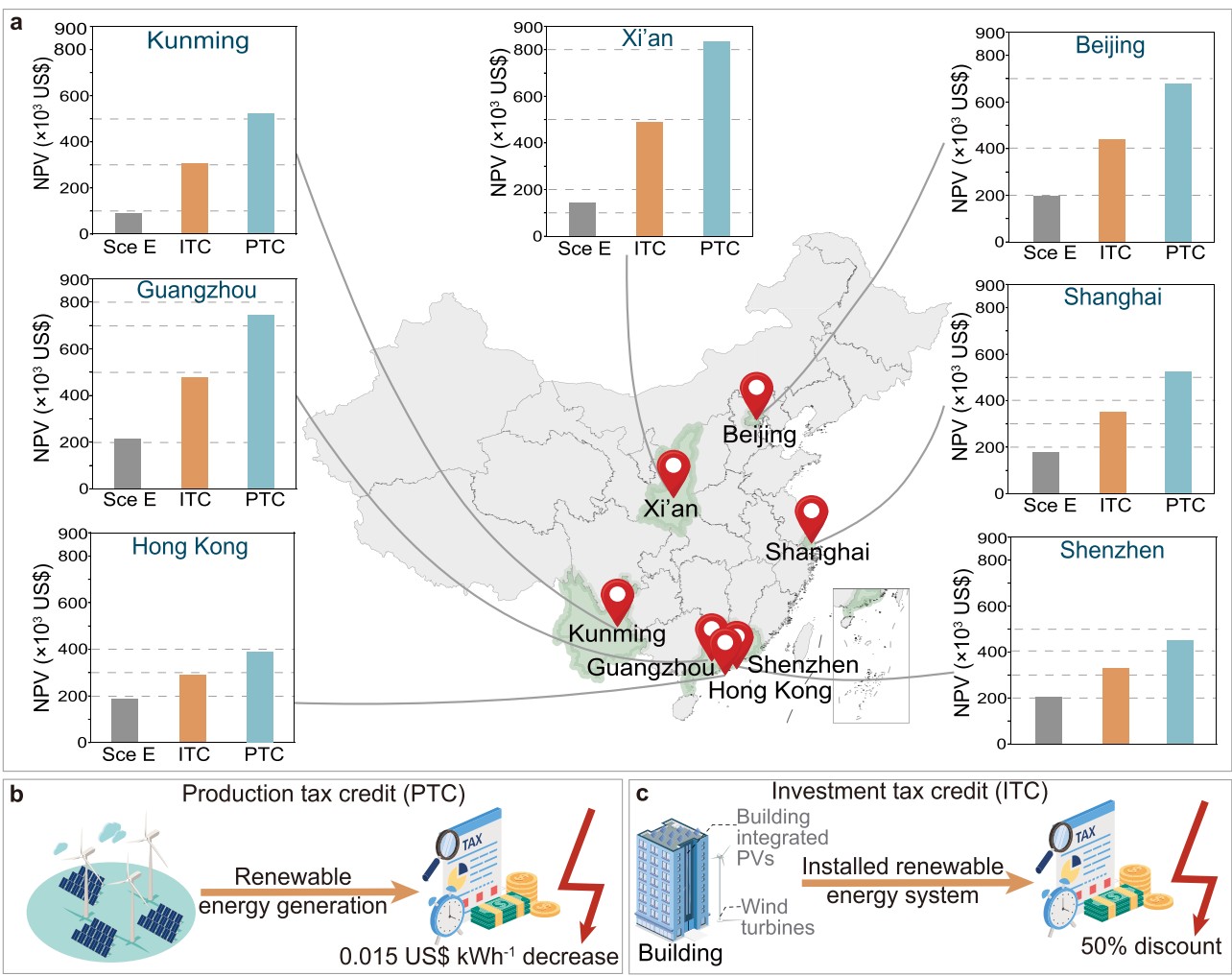

**Fig. 9 | The implementation of production tax credit (PTC) and investment tax credit (ITC). a** Net present values (NPVs) after implementing PTC and ITC; **b** The implementation of PTC; **c** The implementation of ITC.

Under the traditional coal power plant-supported power grid, the replacement of internal combustion engine vehicles by electric vehicles cannot effectively reduce lifecycle carbon emissions unless the operational carbon can be partially offset by cleaner energy with a clean power grid upgrade or renewable energy integration. In terms of lifecycle battery carbon intensity, the operational stage accounts for over half of the total battery carbon intensity in the grid-to-vehicle charging supported by traditional coal-based power plants. The roles of EVs under the traditional grid-to-vehicle charging mode only transfer the emission from traditional fuel vehicles to traditional coal-based power plants, without fundamentally reducing the lifecycle carbon, but increasing the carbon emission instead and leading to environmental issues considering the retirement of batteries.

Battery circular economy with renewable energy sources integration plays essential roles in lowering the battery carbon intensity, and lifecycle zero-carbon batteries can be achieved with embodied carbon (i.e., raw materials, manufacture and assembling processes) completely offset by operational carbon in most climate types through the combination of multi-direction V2X with the cascade reuse strategy on retired EV battery. The cascade EV battery reuse technology can effectively improve the NPV with economic feasibility through the extension in battery service lifetime, while multi-direction V2X interaction exerts a high impact on the battery carbon intensity. Through multi-directional V2X interaction, battery cascade utilization and

lifecycle battery circular economy proposed in this study, the zero-carbon and negative-carbon can be achieved in the battery industry with economic feasibility. The methods proposed in this study are applicable in different climate regions, especially in Europe where electricity prices are high. In addition, through policies such as production tax credit (PTC) and investment tax credit (ITC), investors' willingness can be promoted to achieve carbon neutrality in the batteries, buildings, electric vehicles, power grid and other fields.

The conclusions of this paper provide an effective lifecycle battery carbon-neutral path, which can provide useful suggestions for decentralization, renewable and sustainability, E-mobility with multi-directional V2X interactions, lifecycle battery circular economy and clean power grid transformation. However, there are still limitations in this paper. Firstly, the impact of the energy transformation from traditional grids to clean power grids has not been studied in depth, considering the gradual increase in renewable proportions and economic investments in the local power grid. Secondly, this research is based on the distributed energy system from buildings, rather than the centralized energy system along the coastal lines (such as floating PV farms and off-shore wind turbines). The comparison between distributed energy systems and centralized energy systems, and the participation of EVs in centralized energy systems are also worth studying. Therefore, future work will focus on 1) the carbon emission factor of the local power grid towards clean power grid transformation with gradual change in local energy structures; and 2) centralized

renewable systems along the coastal lines for lifecycle battery sustainability and zero-carbon community.

## Methods
This paper develops six calculation models based on the software TRNSYS (Transient System Simulation). 1) Battery degradation model: simulate the dynamic degradation of the battery under multi-directional V2X interactions. 2) EV-based building-to-building energy sharing model: simulate the energy sharing and interaction among renewable systems, EVs and buildings within the district community. 3) Cascade EV battery utilization model: simulate EV battery replacement and reuse of retired batteries. 4) Battery carbon intensity calculation model: calculate the environmental impact of batteries over the lifecycle. 5) NPV calculation models: calculate the system's economic performance over the lifecycle. 6) EV/ICEV lifecycle carbon intensity calculation model: calculate the lifecycle carbon intensity of electric vehicles (EVs) and internal combustion engine vehicles (ICEVs).

### Battery degradation model
The battery model used in this paper is extended from a previously developed battery model[29]. Supplementary Fig. 9 shows the degradation curve of the battery model, which is characterized by the different degradation speeds of the battery under different depths of discharge (DoD). According to this characteristic, a system based on the number of cycles and dynamic DoD is developed. The dynamic DoD in lithium-iron-phosphate batteries can be calculated by Eq. (1):

$$DoD = FSOC_{\text{peak,cycle}\,n} - FSOC_{\text{valley,cycle}\,n} \tag{1}$$

where $FSOC_{\text{peak,cycle}\,n}$ and $FSOC_{\text{valley,cycle}\,n}$ refer to the highest and the lowest point of $FSOC$ at the $n^{\text{th}}$ cycle.

In addition, according to Supplementary Fig. 9, relative capacity (RC) is used to illustrate the degree of degradation of lithium-ion batteries. The Li-ion battery degradation curve can be polynomial fitted with exponential power at 3 based on the number of cycles (CycleNum) and relative capacity (RC) according to the data in ref. 30, which is expressed in Eq. (2). The values of $k_1$, $k_2$, $k_3$, and $k_4$ under different DoDs are shown in Supplementary Table 2.

$$RC_{DoD} = k_1 CycleNum^3 + k_2 CycleNum^2 + k_3 CycleNum + k_4 \tag{2}$$

### E-mobility based building-to-building energy sharing model
The model develops a network of energy interaction between different types of buildings via electric vehicles[6]. The model includes three types of buildings, a high-rise office building, a high-rise hotel, and 30 residential buildings. The model is equipped with BIPVs, solar thermal collectors and on-site wind turbines for each building. EVs can be charged or discharged from the building's on-site renewable energy to make up for the power shortage. G2V represents the mandatory charging of EVs. Building information and energy demand simulation parameters are shown in Supplementary Note 4. Climate information for simulation is shown in Supplementary Fig. 10a–t and Supplementary Note 5. The parameters and configurations of renewable energy systems are shown in Supplementary Tables 3–7, Supplementary Fig. 11 and Supplementary Note 6. The electricity price is shown in Supplementary Tables 8–22 and Supplementary Note 7.

### Cascade EV battery utilization model
The Cascade EV battery utilization model is used to simulate the renewable energy-EV-grid interactions, reused battery operation, battery SOC and battery relative capacity during the EV battery reuse phase. The reuse of batteries can be divided into three steps, replacement (Fig. 10b), reuse (Fig. 10c) and recycling (Fig. 10d). These three steps occur at different stages of the battery lifecycle, and their

occurrence time points are shown in the battery degradation curves shown in Fig. 10a.

### Battery carbon intensity calculation model
To quantify the carbon emissions of batteries throughout the lifecycle, the carbon intensity (CI) has been proposed as shown by Eq. (3):

$$CI = \frac{CE_{\text{tot}}}{Cap_{\text{sys}}} \tag{3}$$

where the total carbon emission ($CE_{\text{tot}}$) is as calculated by Eq. (4):

$$CE_{\text{tot}} = CE_{\text{raw-manu}} + CE_{\text{ope}} + CE_{\text{recyc}} + CE_{\text{RE}} \tag{4}$$

where $CE_{\text{raw-manu}}$, $CE_{\text{ope}}$, $CE_{\text{recyc}}$ and $CE_{\text{RE}}$ refer to the carbon emission at the raw materials and manufacturing phase, operation phase, carbon emission of the recycling phase, embodied carbon emission of renewable energy systems. The specific calculations of $CE_{\text{raw-manu}}$, $CE_{\text{ope}}$, $CE_{\text{recyc}}$ and $CE_{\text{RE}}$ are as follows:

I) Raw materials and manufacturing phase. Carbon emission factor (CEF) is applied for carbon emission quantification with $CEF_{\text{raw}}$ at 51 kg $CO_{2,e}$ kWh⁻¹ from the raw materials, $CEF_{\text{manu}}$ and $CEF_{\text{remanu}}$ at 34 kg $CO_{2,e}$ kWh⁻¹ from the manufacturing and remanufacturing process, respectively[17]. Therefore, the carbon emission ($CE_{\text{raw-manu}}$) at the raw materials and manufacturing phase can be expressed by Eq. (5):

$$CE_{\text{raw-manu}} = CE_{\text{raw}} + CE_{\text{manu}} + CE_{\text{remanu}} \tag{5}$$

where

$$CE_{\text{raw}} = \left(Cap_{\text{ini},i} + Cap_{\text{repl,new},i} - Cap_{t_{\text{end}},\text{new},i} \cdot \frac{RC_{t_{\text{end}},\text{new},i} - RC_{\text{recyc},i}}{1 - RC_{\text{recyc},i}}\right) \times CEF_{\text{raw}}$$

$$CE_{\text{manu}} = \left(Cap_{\text{ini},i} + Cap_{\text{repl,new},i} - Cap_{t_{\text{end}},\text{new},i} \cdot \frac{RC_{t_{\text{end}},\text{new},i} - RC_{\text{recyc},i}}{1 - RC_{\text{recyc},i}}\right) \times CEF_{\text{manu}}$$

$$CE_{\text{remanu}} = \left(Cap_{\text{remanu},i} - Cap_{t_{\text{end}},\text{remanu},i} \cdot \frac{RC_{t_{\text{end}},\text{remanu},i} - RC_{\text{recyc},i}}{1 - RC_{\text{recyc},i}}\right) \times CEF_{\text{remanu}}$$

where $Cap_{\text{ini},i}$ and $Cap_{\text{repl},i}$ refer to the initial capacity and replacement capacity of $i$th battery, respectively. $Cap_{\text{remanu},i}$ refers to the total capacity of batteries remanufactured from recycled batteries. Considering that at the end of the simulation, there are still EV batteries and reused batteries with remaining relative capacity above the recycling standard ($RC_{\text{recyc},i}$), the carbon emission needs to be deducted to avoid the underestimation. $Cap_{t_{\text{end}},i}$ and $RC_{t_{\text{end}},i}$ refers to the battery capacity and the corresponding relative capacity of $i$th battery at the end of the simulation (i.e., 20 years). $RC_{\text{recyc},i}$ refers to the relative capacity that the $i$th battery needs to be recycled (i.e., $RC$ at either 80% or 60%). The subscript $new$ of $Cap_{\text{repl},i}$ $Cap_{t_{\text{end}},i}$, $RC_{t_{\text{end}},i}$ and $RC_{\text{recyc},i}$ means that the $i$th battery at the end of the simulation is purchasing from the market.

II) Operation phase. During the operation phase, the decarbonization amount by storing excess renewable energy in the battery is calculated by Eq. (6):

$$CE_{\text{ope}} = \sum_{i=1}^{j} \int_0^{t_{\text{end}}} [P_{\text{grid,ch},i}(t) - P_{\text{RE,ch},i}(t)] \cdot CEF_{\text{coal}} dt \tag{6}$$

where $CE_{\text{ope}}$ refers to the equivalent carbon emission of the operation phase. $P_{\text{grid,ch},i}$ and $P_{\text{RE,ch},i}$ refer to the grid power and renewable energy charged to the $i$th battery. $CEF_{\text{coal}}$ is the average $CO_2$ emission factor of the coal power in Beijing at 1.0029 kg $CO_{2,e}$ kWh⁻¹, in Guangzhou and

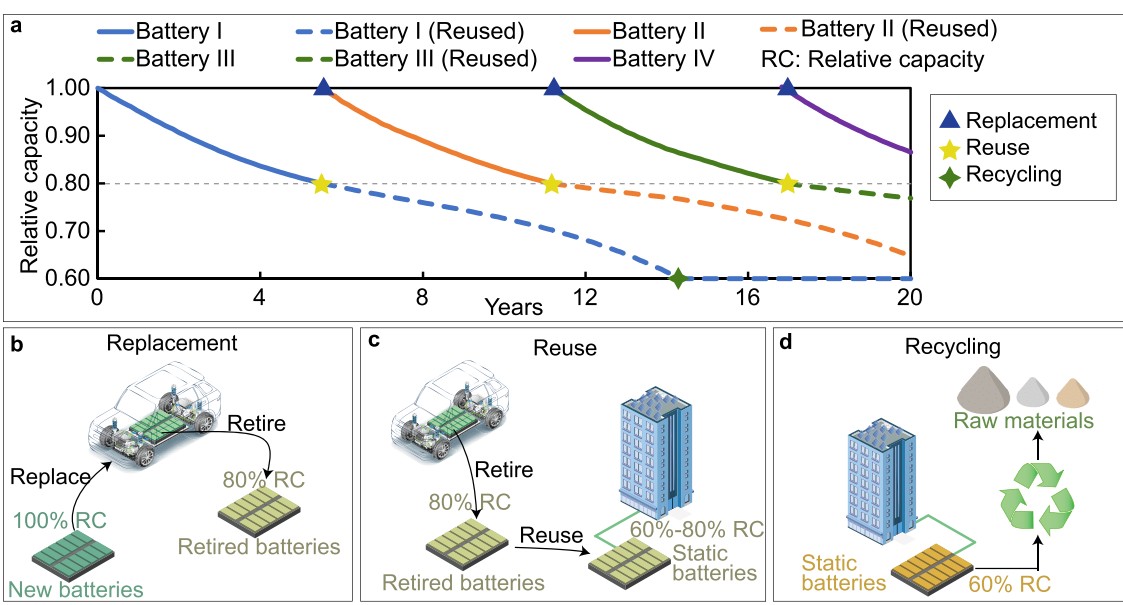

**Fig. 10 | Primary and second-life batteries with relative capacity. a** Degradation curves of primary and second-life batteries; **b** Electric vehicle battery replacement phase; **c** The process of EV battery reuse phase; **d** The recycling of static batteries.

Shenzhen at 0.8652 kg $CO_{2,e}$ kWh$^{-1}$, in Hong Kong at 0.9681 kg $CO_{2,e}$ kWh$^{-1}$, in Xi'an at 0.9532 kg $CO_{2,e}$ kWh$^{-1}$, in New York at 1.0433 kg $CO_{2,e}$ kWh$^{-1}$ in Berlin at 0.8670 kg $CO_{2,e}$ kWh$^{-1}$, and in Singapore at 0.9120 kg $CO_{2,e}$ kWh$^{-1}$, respectively.

III) Recycling phase. In the recycling phase, the carbon emission factor of recycling ($CEF_e$) is 69.8 kg $CO_{2,e}$ kWh$^{-1}$ for Li-ion batteries[17]. The carbon emission ($CE_{recyc}$) at the recycling phase can be expressed by Eq. (7):

$$CE_{recyc} = \sum_{i=1}^{j} Cap_{recyc,i} \cdot CEF_{recyc} \tag{7}$$

where $Cap_{recyc,i}$ refer to the recycled battery capacity of the $i$th battery.

IV) Embodied carbon of renewable energy. The embodied carbon of renewable energy equipment can be defined as the carbon emission generated during manufacturing, transportation, operation, maintenance and recycling of renewable energy equipment.

For PVs, the embodied carbon emission per kWp (kilowatts peak power) is shown in Eq. (8)[31]:

$$ECE_{PV,unit} = 560 \text{ kg } CO_{2,e} \text{ kWp}^{-1} \tag{8}$$

For wind turbine, the embodied carbon per kW can be calculated by Eq. (9):

$$ECE_{WT,unit} = 1959.2 \times P_{WT,rated}^{-0.224} \tag{9}$$

where $ECE_{WT, unit}$ refers to the embodied carbon emission of the wind turbine with a unit of kg $CO_{2,e}$ kW$^{-1}$, and $P_{WT, rated}$ refers to the rated power of the wind turbine.

Equation (9) is obtained by fitting as shown in Supplementary Fig. 12. The points shown in Supplementary Fig. 12 represent wind turbines with corresponding rated power and their embodied carbon emission with the data from ref. 32–38.

Therefore, the carbon emission from building integrated renewable systems, $CE_{RE}$, is calculated in Eq. (10):

$$CE_{RE,ch} = \frac{E_{RE,ch}}{E_{RE}} \times \left( P_{PV} \cdot ECE_{PV,unit} + \sum_{i=1}^{j} P_{rated,i} \cdot ECE_{WT,unit,i} \right) \tag{10}$$

where $E_{RE,ch}$ and $E_{RE}$ refer to the renewable energy charged to the battery and the renewable energy generated by PVs and wind turbines. $P_{PV}$ is the peak power of the PV panel at the standard condition. $ECE_{PV,unit}$ refers to the embodied carbon emission of PVs per kWp. $ECE_{WT,unit}$ refers to the embodied carbon emission of the wind turbine per kW, and $P_{WT,rated}$ refers to the rated power of the wind turbine.

In addition, in order to quantify embodied carbon emissions of renewable systems, the embodied carbon emission factor of renewable systems ($ECEF_{RE}$) is introduced to quantify the carbon emitted by renewable systems for generating a unit of electricity during its lifecycle. The calculation is described in Eqs. (11, 12):

$$ECEF_{RE} = \frac{ECE_{RE}}{E_{RE}} \tag{11}$$

$$E_{RE} = \int_{t=0}^{t=lifetime} P_{RE} \, dt \tag{12}$$

where $ECE_{RE}$ is the embodied carbon emission of solar PV, BIPVs or wind turbines during their lifetime processes, which is calculated by Eqs. (8, 9) based on embodied carbon emission per kWp, involving manufacturing, transportation, operation, maintenance and recycling processes. $E_{RE}$ is the total renewable generation of solar PV, BIPVs or wind turbines throughout their lifecycle processes. $P_{RE}$ is the power of renewable generation. In addition, the lifetime for PVs and wind turbines is 25 years and 20 years, respectively.

V) Total battery capacity within the system boundary. The total battery capacity of the integrated multi-energy system is dependent on the energy boundary and will be different between the case without and with the second-life battery reuse strategy. The total capacity of the integrated multi-energy system is calculated as:

$$Cap_{sys} = \begin{cases} \sum_{i=1}^{j} Cap_{ini,i} & \text{without second life battery reuse strategy} \\ \sum_{i=1}^{j} (Cap_{ini,i} + Cap_{repl,new,i}) & \text{with second life battery reuse strategy} \end{cases} \tag{13}$$

## NPV calculation model

In this paper, net present value (*NPV*) refers to the 20-year revenue of the entire system, and its calculation equation is:

$$NPV = \Delta C_{imp,save} - \Delta C_{recyc} - \Delta C_{remanu} - \Delta C_{repl,new} - \Delta C_{repur} \\ - \Delta C_{O\&M} - \Delta IC_{RE} - \Delta IC_{bat} \tag{14}$$

The specific calculation of Eq. (14) is listed in Supplementary Note 8. In terms of the Supplementary Note 8, regarding Supplementary Equation (1), $\Delta C_{imp,save}$ refers to the grid electricity import cost saved when renewable energy is used to replace grid electricity, and the electricity price is shown in Supplementary Tables 8–22 and Supplementary Note 7. $\Delta C_{recyc}$ in Supplementary Equation (2) refers to the cost of recycling EV batteries or second-life batteries at the end of their life (57 \$ kWh$^{-1}$)[39]. $\Delta C_{remanu}$ in Supplementary Equation (3) refers to the cost of remanufacturing new batteries using recycled battery materials (40 \$ kWh$^{-1}$)[40]. $\Delta C_{repl,new}$ in Supplementary Equation (4) refers to the cost of replacing old batteries with new ones in EVs (125 \$ kWh$^{-1}$)[24]. $\Delta C_{repur}$ in Supplementary Equation (5) refers to the repurposing cost required to reuse retired EV batteries as second-life batteries (35 \$ kWh$^{-1}$)[23]. $\Delta C_{O\&M}$ in Supplementary Equation (6) refers to the operation and maintenance costs of L-ion batteries, Solar PVs, BIPVs, and wind turbines in the system. For Solar PVs, BIPVs and wind turbines, the annual $C_{O\&M}$ is 5% of the initial cost. For Li-ion batteries, the annual $C_{O\&M}$ is 0.5% of the initial cost. $\Delta IC_{RE}$ in Supplementary Equation (7) refers to the initial investment cost of Solar PVs, BIPVs and wind turbines used. All "Δ" refer to the result calculated as the difference between the studied case and the reference case. Note that if the secondary-life battery reuse strategy is not adopted, $\Delta C_{repl,new}$ and $\Delta C_{repur}$ will be 0 due to the lack of purchase of new batteries and repurposing of retired batteries. $\Delta IC_{bat}$ refers to the initial investment cost of Li-ion batteries and will be 0 since neither the scenarios nor the paradigms increase battery usage.

## Lifecycle carbon emission calculation model for vehicles

The calculation of carbon emissions for vehicles is similar to batteries and can be divided into four phases: raw materials, manufacturing, operation, and recycling. This paper calculates the lifecycle carbon emission of electric vehicles (EVs) and internal combustion engine vehicles (ICEVs), respectively, as shown in Eq. (15)

$$CE_{vehicle} = \frac{CE_{tot,vehicle}}{M_{lifecycle}} \tag{15}$$

where $CE_{vehicle}$ is the lifecycle carbon emission for vehicles. $CE_{tot,vehicle}$ is the total carbon emission for vehicles in their lifecycle process. $M_{lifecycle}$ is the mileage of vehicles in their lifecycle, assumed to be 300,000 km in this research[41]. In addition, $CE_{tot,vehicle}$ is calculated as shown in Eq. (16):

$$CE_{tot,vehicle} = CE_{raw,vehicle} + CE_{manu,vehicle} + CE_{ope,vehicle} + CE_{recyc,vehicle} \tag{16}$$

where $CE_{raw,vehicle}$, $CE_{manu,vehicle}$ and $CE_{recyc,vehicle}$ refer to the carbon emission of the raw materials, vehicle manufacturing and recycling, respectively. $CE_{ope,vehicle}$ refers to the carbon emission during the vehicle traveling. $CE_{raw,vehicle}$, $CE_{manu,vehicle}$ and $CE_{recyc,vehicle}$ both include two parts, vehicle body and batteries (only for EVs). The calculation of battery carbon emission is based on Eqs. (5) and (7). The embodied carbon emission data of the vehicle body is cited from the research by Qiao et al.[42]. In addition, $CE_{ope,vehicle}$ is calculated based on Eqs. (6) and (17) for EVs and ICEVs, respectively.

$$CE_{ope,ICEV} = M_{lifecycle} \times E_{L\,km^{-1}} \times ECEF_{fuel} \tag{17}$$

where $E_{L\,km^{-1}}$ is the fuel consumption of ICEVs, and $ECEF_{fuel}$ refers to the equivalent carbon emission of fuel, including gasoline (2.291 g $CO_{2,e}$ L$^{-1}$, consumed by ICE-cars) and diesel fuel (2.729 g $CO_{2,e}$ L$^{-1}$, consumed by ICE-buses)[43]. The parameters required for calculation are shown in Supplementary Tables 23–26, Supplementary Fig. 13 and Supplementary Note 9.

## Data availability

The data used as model inputs are provided in the Supplementary Information file. The relevant raw data from each figure generated in this study are provided in the Source Data file. The data containing the results of all calculations generated in this study have been deposited in the Zenodo database under the accession code https://doi.org/10.5281/zenodo.10553509. Source data are provided in this paper.

## Code availability

TRNSYS code and models used to generate the results reported in this study are available from the corresponding author upon request.

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

## Acknowledgements

This work was supported by the National Development and Reform Commission (2023-Dual Carbon-3, Y.Z.), Natural Science Foundation Project (General Project)-Guangdong Basic and Applied Basic Research Fund (2414050003253, Y.Z.), Regional joint fund youth fund project (2022A1515110364, P00038-1002, Y.Z.), Guangdong Basic and Applied Basic Research Foundation 2023 (2023A04J1035, P00121-1003, Y.Z.), Joint Funding of Institutes and Enterprises in 2023 (2023A03J0104, P00054-1003,1004, Y.Z.), Green Tech Fund in the Hong Kong Special Administrative Region 'Developing low-cost PEM electrolysis at scale by optimizing transport components and electrode interfaces' (GTF202220034, Y.Z.). HKUST(GZ)-enterprise cooperation project (R00017-2001, Y.Z.), HKUST(GZ)-enterprise cooperation project 'Research on Development of Vehicle-City-Network and Electric Vehicle Charging Pile Industry' (R00114-2001, Y.Z.). HKUST(GZ)-enterprise cooperation project (R00017-2001, Y.Z.), HKUST(GZ)-enterprise cooperation project 'Optimization Design of Proton Exchange Membrane Fuel Cell Plate' (R00072-2001, Y.Z.), HKUST(GZ)-enterprise cooperation project 'Next-generation radiant cooling for the built environment' (R00079-2001, Y.Z.). This research is supported by The Hong Kong University of Science and Technology (Guangzhou) startup grant (G0101000059, Y.Z.). This work was also supported in part by the Project of Hetao Shenzhen-Hong Kong Science and Technology Innovation Cooperation Zone (HZQB-KCZYB-2020083, Y.Z.).

## Author contributions

A.S. and Y.Z. designed the study; A.S. performed the data analysis; Y.Z. and S.Z. provided supervision. A.S. and Z.D. wrote the paper; S.Z. provided comments. A.S., Z.D., S.Z. and Y.Z. provide revision.

## Competing interests

The authors declare no competing interests.
