## [Peer Review File · Nature Communications]

REVIEWER COMMENTS

Reviewer #1 (Remarks to the Author):

Under the traditional coal power plant-supported power grid, the replacement of internal combustion engine vehicles (ICE-cars and ICE-buses) by electric vehicles (E-cars and E-buses) cannot effectively reduce lifecycle carbon emissions, unless the operational carbon can be partially offset by cleaner energy with clean power grid upgrade or renewable energy integration. This paper provides an effective lifecycle battery carbon-neutral path, which can provide useful suggestions for decentralization, renewable and sustainability, E-mobility with multi-directional V2X interactions, lifecycle battery circular economy, and clean power grid transformation. Net present value with battery replacement frequencies, battery cycling aging, and electricity policies is calculated to guide the economic feasibility of energy, social, and governance investment behaviors. There is no issue with this article. And their future work is of great significance to the development of society.

Reviewer #2 (Remarks to the Author):

Review Comments for LCA Battery SimuPlat: Lifecycle battery carbon intensity quantification and climate-adaptive E-mobility circular economy for carbon-neutrality in China

General and In Summary and Introduction sections

1. The reviewer is of the opinion that although the conclusions of this paper seem pretty obvious and although not trivial and not insignificant, but still very much intuitive, nevertheless, the analytical approaches developed in this paper, still deserve merit to be considered a novel research contribution to the body of knowledge.

2. The reviewer is curious to see if the authors can provide a general perspective of the global setting by extending the present analysis for China and commenting on how these factors will play out. The reviewer recommends to add a separate section for this purpose.

B. In section Carbon emission factor on various renewable systems across different climates

1. The reviewer is interested at knowing and recommends that the authors also address the applicability and analysis of equations (1)-(3) for other renewable sources beyond solar and wind, like geothermal, hydro, ocean tidal etc.

C. In section EV battery for transportation with renewable energy

1. The reviewer is curious about the status of production tax credit (PTC) and investment tax credit (ITC) in the regions that the authors have done the analysis and whether or not the authors have taken those into account in this study. If not, the reviewer recommends to do so and state that, after taking those into account, how's this scenario as compared to the other scenarios in terms of expensiveness.

Reviewer #3 (Remarks to the Author):

The study explores a hypothetical future vision of a localized energy system with residential and office buildings, EV charging on-site generation (through wind and solar). The goal is to calculate a representative life cycle carbon intensity for the batteries (not specified but assumed to be lithium ion batteries), as well as an economic assessment of net present value. A variety of scenarios are explored beyond a reference (business as usual) case, such as multi-directional vehicle to building/building to vehicle charging; reuse of EV batteries in a second life as stationary building energy storage, and a combination thereof. The results suggest that specific sets of scenarios, locations, and input values can result in a battery life cycle that is net-zero carbon. The combination of individual models and series of equations encompassing EV, energy storage, and

generation operations; battery degradation, transient building – vehicle interactions; and battery reuse operations is unique and of significance to the field. This linked approach has been missing from the established literature and is clearly needed as governments around the world pursue energy transition strategies to decarbonize electricity and associated technologies. I appreciate the vision necessary to create these links and produce the work, which was complex to manage.

Most of the components of the methodology that are described in the paper appear appropriate, with a few specific exceptions/questions mentioned below. Following from that, the straightforward reference case and renewables case provide the expected results which are clearly outlined in the conclusions. However, the paper does not successfully justify the results and conclusions in other places because of the lack of explanation of the methodology used. For example, while many input values are provided in the appendix, it is difficult to understand the validity of some results and confirm the interpretation without understanding the operational profiles of the equipment in the scenarios. This also means there is insufficient detail provided for the work to be reproduced. Additional documentation of the methodology and how it led to the stated conclusions is needed. Therefore, it is recommended that a major revision and re-review be completed before publication is considered.

Another major feature that is missing from the paper is an early statement of the fact this is an exploratory vision of the future. It uses a variety of assumptions, many of which are quite liberal or beyond common ranges used by others, with no sourcing or explanation of how they were chosen. For example, CEF values for raw materials extraction are on the low end of ranges published by others for China (e.g., Xu et al., for US, China and EU have 40-80%, and this paper chose 51 kg CO_{2e}/kWh). If those values were chosen to maximize the possibility of achieving the “net zero carbon” result, that is fine, but it should be clearly stated in the introductory material and the interpretation/conclusions, and not left for the reader to determine.

Additionally, there are sections of the report that are difficult to interpret, and this is likely due to the use of uncommon English phrases or vocabulary. As a result, reading the document to simply understand the intent of the statements took a substantial amount of time, without evaluating the technical quality of the work. Suggest that a native English speaker work with the authors to further refine the language for clarity. This will assist in future interpretation of the validity of the results.

-

Specific Comments:

- The V2B/B2V aspect of the paper is a particularly novel component, and yet there is no information provided on how this interaction works, what type of daily schedule it operates, how that affects depth of discharge of the batteries (which will affect degradation), etc. All that is stated is a general “multi-directional charging” statement, along with the results. The process and associated calculations must be explained in the paper, and referred to.
- What is the charge/discharge cycle and depth of discharge for the EVs in the various scenarios? How might that affect the degradation, and then the carbon intensity?
- Top of page 8 statement that shuttle buses are “much lower” life cycle carbon emissions is overstated given the 10% difference
- Statements about the CER in the middle of page 8 discussion decreases in various regions, but not the time frame over which they are relevant or if they are part of the future hypothetical case study. If they are future values, are they aspirational or are there policy targets in place to achieve them
- Transportation of the renewable and battery systems was not included in the life cycle calculations. This is an important element to include. Perhaps it was not because the expectation is transport short distances within China due to active lithium ion battery industry there. Again, the choice to eliminate this life cycle phase must be addressed, and justified.
- In equations 2 the carbon emissions of recycling (or reuse) are included. It is unclear why, as this would normally be classified separately from materials processing and manufacture. Similarly, equation 12 in the appendix includes remanufacturing in the manufacturing calculation. There is an approach in the life cycle allocation of materials and outcomes during reuse of a product that assigns a fraction to the first life and a fraction to the second life. However, this is not explained at all in the paper and must be.
- Equation 2- what is the difference between manufacture and production?
- Page 11 states that zero-energy approach for the localized grid is chosen; that the total annual

generation matches the annual electricity consumption. Is this realistic given today's technologies or aspirational? If aspirational, it may account for what appear to be oversized renewables in Tables S22-24. For example, the US Department of Energy describes small scale wind turbines for residences in the range of 400W to 20kW; the section in this paper is the maximum value. Similarly, hotels are expected to range from 1-100kW size, but this paper uses 500kW.

- Page 11 – why is scenario 2 pure renewables a solar scenario, when it is clear from the other data and figures in the paper that wind is a much more common and valued resource in China?
- Page 13 – the paper notes “electricity prices are generally higher than the costs of solar power”. This is an unusual statement from the point of view of certain countries and grid design. Can the reasoning behind this be explained?
- Page 13-14 – sentence does not make sense “...the overwhelmed dominance of the import cost savings over the battery cycling ageing costs”. Does this intend to state the degradation is slower in this scenario? What is an “import cost savings”?
- Page 14- why is the cost of the reused battery higher?
- Some graphics have acronyms that are not defined (e.g., ST, RC). Please define them.
- Figure 6 – I am confused how the life cycle carbon intensity goes negative in panel B. Should it not hold constant rather than increase?
- Page 22 – the battery degradation data used was based on a first life situation, which is fine. However, the paper does not state what degradation data was used for the second life / reuse situation.

Detailed reply to the reviewers

1st Revision (Feb 2024)

Specific reply to the comments one by one:

Reviewer #1 (Remarks to the Author): Under the traditional coal power plant-supported power grid, the replacement of internal combustion engine vehicles (ICE-cars and ICE-buses) by electric vehicles (E-cars and E-buses) cannot effectively reduce lifecycle carbon emissions, unless the operational carbon can be partially offset by cleaner energy with clean power grid upgrade or renewable energy integration. This paper provides an effective lifecycle battery carbon-neutral path, which can provide useful suggestions for decentralization, renewable and sustainability, E-mobility with multi-directional V2X interactions, lifecycle BATTERY LIFE economy, and clean power grid transformation. Net present value with battery replacement frequencies, battery cycling aging, and electricity policies is calculated to guide the economic feasibility of energy, social, and governance investment behaviors. There is no issue with this article. And their future work is of great significance to the development of society.

Reply: Thank you for your recognition and praise of our article. We are glad to provide an effective lifecycle battery carbon-neutral path to promote decentralization, renewable and sustainable development, e-mobility with multi-directional V2X interactions, lifecycle battery circular economy, and clean power transformation. We will continue to work in this field to further improve our research and contribute to the development of society. Once again, thank you for your support, evaluation and suggestions.

Reviewer #2 (Remarks to the Author): Review Comments for LCA Battery SimuPlat: Lifecycle battery carbon intensity quantification and climate-adaptive E-mobility circular economy for carbon-neutrality in China

General and In Summary and Introduction sections

1. The reviewer is of the opinion that although the conclusions of this paper seem pretty obvious and although not trivial and not insignificant, but still very much intuitive, nevertheless, the analytical approaches developed in this paper, still deserve merit to be considered a novel research contribution to the body of knowledge.

Reply: Thank you very much for your precious time and professional comment. Based on your comments, we have added the Supplementary Note 1 to highlight the analysis method of this article, which is described below:

Supplementary Figure 1 The methodology flow chart of this study.

The main feature of our method is that it is based on multiple models and is organically combined and used in series based on modular design. As shown in the Supplementary Fig. 1, firstly, we obtained climate database, renewable energy data, vehicle operation data, and electricity price data for different climate zones. Based on these data, considering different energy interaction modes, battery reuse strategies and relative relationship between renewable energy supply and energy demand, we reasonably preset 5 scenarios (Scenario A-E) and 3 paradigms (Paradigm A-C). We then established corresponding building-vehicle energy sharing network models and their analytical models, including building thermodynamics models, integrated building energy systems (e.g., building services systems, rooftop PVs, BIPVs, solar thermal collectors and so on), building-vehicle interaction and battery degradation models. We then analysed the following three projects: life cycle carbon emissions comparison between electric vehicles and internal combustion engine vehicles, calculation of carbon emission factors for renewable energy, and quantifications of battery life cycle carbon intensity, especially considering the complexity in the operational stage. Based on the analysis results, we finally proposed zero-carbon pathways for battery life cycles in different climate zones.

If there is any more comment, please suggest them, and we will be very glad to make further improvements for this paper draft.

2. The reviewer is curious to see if the authors can provide a general perspective of the global setting by extending the present analysis for China and commenting on how these factors will play out. The reviewer recommends to add a separate section for this purpose.

Reply: Thank you very much for your very professional comment. In this Revision R1, according to your comments, we add a new subsection ‘*The general perspective of global setting*’. To explore whether our proposed methods and zero-carbon pathways are applicable globally, we selected three cities in the United States, Europe and Singapore for analysis, considering that EVs and net-zero energy buildings have been promoted and adopted in these regions. We obtained the results as shown in the figure below, and added a section to the main text for result display, explanation and analysis. The following is the content in the main text:

Fig. 8 Lifecycle battery carbon intensity and net present values (NPVs) in New York, Berlin and Singapore.

‘The multiple methods proposed in this article for battery carbon neutrality transformation were further verified and promoted from a global perspective. New York, Berlin and Singapore from different continents are selected. As shown in Fig. 8, under the traditional scenario (Scenario A), the emissions of the three cities are relatively high (1925.17, 1628.05, and 1703.89 kg CO_{2,e}/kWh for New York, Berlin and Singapore, respectively). Among them, the data of Singapore are close to Chinese cities, while those of New York and Berlin are much lower. The main reason is that the energy demand of buildings in the United States and Europe is much lower due to the more moderate weather climate. In the ideal scenario (Scenario B), all cities can achieve negative carbon intensity. Furthermore, with the implementation of strategies (such as multi-direction V2X interaction and battery cascade utilisation), carbon intensity can be significantly reduced compared to the Scenario A, as shown in Scenario C and Scenario D. The Scenario E with simultaneous adoptions of two strategies (multi-direction V2X interaction and battery cascade utilisation) achieved carbon intensity close to the ideal scenario.

In terms of NPV, based on the results of Scenarios C, D and E, it can be concluded that both V2X interaction and battery cascade utilization are profitable. However, it is noted that the NPV of Berlin is

much higher than that of other cities. This is because the electricity prices in Berlin are much higher (0.7 US\$/kWh) than (average at 0.216 US\$ kWh⁻¹) or Singapore (average at 0.200 US\$ kWh⁻¹), which makes the import cost saving in Scenarios B, C, D and E very high, resulting in a high NPV.

In general, the zero-carbon path and its analysis method represented by multi-direction V2X interaction, battery cascade utilization and lifecycle circular economy can not only guide the energy transformation of various regions in China, but also provide suggestions to energy and environmental policy makers around the world.' (*Section-The general perspective of global setting*)

If there is any more comment, please suggest them, and we will be very glad to make further improvements for this paper draft.

B. In section Carbon emission factor on various renewable systems across different climates

1. The reviewer is interested at knowing and recommends that the authors also address the applicability and analysis of equations (1)-(3) for other renewable sources beyond solar and wind, like geothermal, hydro, ocean tidal etc?

Reply: Thank you very much for your very professional comment. Other renewable sources beyond solar and wind, like geothermal, hydro, ocean tidal, are important for low-carbon or even carbon neutrality transition. However, compared to solar and wind energy, the geothermal, hydro and ocean tidal are highly geographically dependent. Furthermore, both solar and wind energy can be integrated in a distributed form in buildings, while hydro and ocean tidal are generally used in centralised energy supply. Other forms of renewable energy may not be suitable for distributed energy systems, considering their climate-dependent characteristics, resource availability in different locations, and shortcomings, as shown below.

For geothermal energy: a common application of geothermal energy in distributed energy system is ground source heat pumps (GSHPs). However, it cannot be used for a long time in tropical and some subtropical areas because these areas have long-term cooling demands. Therefore, the use of GSHPs will lead to long-term uneven balance in heating and cooling of the underground, reducing the coefficient of performance of GSHPs.

For hydropower: if hydropower is used to power the distributed energy system, it requires the presence of nearby rivers, which is not applicable in our studied cities.

For the ocean tidal energy: it can only be used in coastal cities with a limited scope of application, and is not very suitable for distributed energy systems.

Based on above considerations, this article does not consider other forms of renewable energy such as geothermal, hydropower and tidal power. The equations (1)-(3) can be used for embodied carbon emission factor calculation of other renewable sources beyond solar and wind, like geothermal, hydro, ocean tidal etc, if necessary. The authors will calculate this in following up studies, when geothermal, hydro, ocean tidal are typical energy sources in the studied cases or regions.

If there is any more comment, please suggest them, and we will be very glad to make further improvements for this paper draft.

C. In section EV battery for transportation with renewable energy

1. The reviewer is curious about the status of production tax credit (PTC) and investment tax credit (ITC) in the regions that the authors have done the analysis and whether or not the authors have taken those into account in this study. If not, the reviewer recommends to do so and state that, after taking those into account, how's this scenario as compared to the other scenarios in terms of expensiveness.

Reply: Thank you very much for your very professional comment. In this revision R1, according to your comments, we add a new subsection '*Impact of tax reduction exemption incentives*' for the discussion. ITC and PTC are indeed important parts of the application and promotion of renewable energy. It is a pity that we did not consider this factor before, so we added a subsection to show the impact on NPV in the presence of policies such as production tax credit (PTC) and investment tax credit (ITC). The following is the text of this subsection:

'Production tax credit (PTC) and investment tax credit (ITC) are two policies announced by the U.S. government to promote clean energy. Among them, PTC stated that taxes can be reduced when clean energy generates electricity, and the amount is 0.015 US\$/kWh (only in the first 10 years). ITC stated that investors can have reduced taxes when investing in PV panels, and the reduced amount is 30% currently. These two policies are mutually exclusive, and investors can only choose one of them.

There are similar policies in China. For example, the Chinese government will exempt photovoltaic and wind power products from 50% of the value-added tax (the rate is 13%). This policy is similar to the ITC. However, China currently does not implement any policy similar to PTC. Therefore, this subsection will also explore the feasibility of implementing similar policies in China based on the U.S. PTC policy, and provide guidance and suggestions for future policy development.

Fig. 9 NPVs after the implementation of production tax credit (PTC) and investment tax credit (ITC).

Based on the above assumptions, the benefits are calculated that can be brought by implementing similar ITC and PTC policies across China, as shown in Fig. 9. In the original Scenario E, the NPV of various places was approximately $90.73\text{--}211.56 \times 10^3$ US\$. After the introduction of the ITC policy, the NPV increased to $287.22\text{--}489.89 \times 10^3$ US\$. After the PTC policy was used, the NPV increased to $389.97\text{--}835.75 \times 10^3$ US\$. It should be noted that due to the high tax exemption limit of the PTC policy, PVs and wind turbines are equivalent to zero tax after the introduction of the PTC policy.

The above results show that using tax reduction or exemption policies can significantly increase investment returns and enhance investors' confidence. However, how to better design PTC-like policies in China requires further discussion. This study can provide frontier guidance and suggestions for future policy development in China or other regions.' (Section-*Impact of tax reduction exemption incentives*) If there are any more comments, please suggest them, and we will be very glad to make further improvements for this paper draft.

Reviewer #3 (Remarks to the Author):

The study explores a hypothetical future vision of a localized energy system with residential and office buildings, EV charging on-site generation (through wind and solar). The goal is to calculate a representative life cycle carbon intensity for the batteries (not specified but assumed to be lithium ion batteries), as well as an economic assessment of net present value. A variety of scenarios are explored beyond a reference (business as usual) case, such as multi-directional vehicle to building/building to

vehicle charging; reuse of EV batteries in a second life as stationary building energy storage, and a combination thereof. The results suggest that specific sets of scenarios, locations, and input values can result in a battery life cycle that is net-zero carbon.

The combination of individual models and series of equations encompassing EV, energy storage, and generation operations; battery degradation, transient building – vehicle interactions; and battery reuse operations is unique and of significance to the field. This linked approach has been missing from the established literature and is clearly needed as governments around the world pursue energy transition strategies to decarbonize electricity and associated technologies. I appreciate the vision necessary to create these links and produce the work, which was complex to manage.

Reply: Thank you very much for taking valuable time to help review our work and your very professional comments. In this Revision R1, we answer all comments one by one. If there is any more comment, please suggest them, and we will be very glad to make further improvements for this paper draft.

Most of the components of the methodology that are described in the paper appear appropriate, with a few specific exceptions/questions mentioned below. Following from that, the straightforward reference case and renewables case provide the expected results which are clearly outlined in the conclusions. However, the paper does not successfully justify the results and conclusions in other places because of the lack of explanation of the methodology used. For example, while many input values are provided in the appendix, it is difficult to understand the validity of some results and confirm the interpretation without understanding the operational profiles of the equipment in the scenarios. This also means there is insufficient detail provided for the work to be reproduced.

Reply: Thank you very much for your professional comments. In this Revision R1, according to your comments, we add more explanations of the methodology. To help readers better reproduce the details of the article, we have added the operational profiles of the equipment in the ‘Supplemental information’, which includes the following content:

- ✧ The operating principle of building-transportation energy sharing network
- ✧ Various parameter information, operating schedules and embodied carbon emissions of EVs
- ✧ Electricity prices in various cities
- ✧ Building operation information
- ✧ Weather information for various cities
- ✧ Renewable energy equipment parameters

For more details, please check the ‘Supplemental information’. Readers can validate research results and confirm the interpretation by inputting the operational profiles of the equipment in the scenarios.

If there is any more comment, please suggest them, and we will be very glad to make further improvements for this paper draft.

Additional documentation of the methodology and how it led to the stated conclusions is needed. Therefore, it is recommended that a major revision and re-review be completed before publication is considered.

Reply: Thank you very much for your very professional comment. In the revision R1, the Supplementary Note 1 is provided to describe the methodology and show the stated conclusions, as shown below.

Supplementary Figure 1 The methodology flow chart of this study.

The main feature of our method is that it is based on multiple models and is organically combined and used in series based on modular design. As shown in the Supplementary Fig. 1, firstly, we obtained climate database, renewable energy data, vehicle operation data, and electricity price data for different climate zones. Based on these data, considering different energy interaction modes, battery reuse strategies and relative relationship between renewable energy supply and energy demand, we reasonably preset 5 scenarios (Scenario A-E) and 3 paradigms (Paradigm A-C). We then established corresponding building-vehicle energy sharing network models and their analytical models, including building thermodynamics models, integrated building energy systems (e.g., building services systems, rooftop PVs, BIPVs, solar thermal collectors and so on), building-vehicle interaction and battery degradation models. We then analysed the following three projects: life cycle carbon emissions comparison between electric vehicles and internal combustion engine vehicles, calculation of carbon emission factors for

renewable energy, and quantifications of battery life cycle carbon intensity, especially considering the complexity in the operational stage. Based on the analysis results, we finally proposed zero-carbon pathways for battery life cycles in different climate zones.

If there is any more comment, please suggest them, and we will be very glad to make further improvements for this paper draft.

Another major feature that is missing from the paper is an early statement of the fact this is an exploratory vision of the future. It uses a variety of assumptions, many of which are quite liberal or beyond common ranges used by others, with no sourcing or explanation of how they were chosen. For example, CEF values for raw materials extraction are on the low end of ranges published by others for China (e.g., Xu et al., for US, China and EU have 40-80%, and this paper chose 51 kg CO_{2e}/kWh). If those values were chosen to maximize the possibility of achieving the “net zero carbon” result, that is fine, but it should be clearly stated in the introductory material and the interpretation/conclusions, and not left for the reader to determine.

Reply: Thank you very much for your very professional comment. In this Revision R1, we add the description in both introductory material and the interpretation/conclusions.

‘Note that, due to the complexity and inconsistency of data sources, the minimum values were chosen to maximize the possibility of achieving the “net zero carbon” result.’ (**INTRODUCTION**)

‘Conservative analysis was conducted with the minimum values to maximize the possibility of achieving the “net zero carbon” result.’ (**CONCLUSION**)

Furthermore, the data used in this article are not only derived from assumptions. Most data are based on reasonable data adopted from existing literature.

For example, the battery price, 125 \$/kWh, is from an estimation of Li-ion battery pack price in 2021. Here is the source: *Fig. 6, Deshwal, D., Sangwan, P., Dahiya, N., 2022. Economic Analysis of Lithium Ion Battery Recycling in India. Wireless Personal Communications 124(4), 3263-3286.* Since Bloomberg bases its forecast **on global Li-ion battery prices, its price is an average value** and has a wide range of applications. The battery recycling cost is from the following reference: *Lander, L., Cleaver, T., Rajaeifar, M.A., Nguyen-Tien, V., Elliott, R.J.R., Heidrich, O., Kendrick, E., Edge, J.S., Offer, G., 2021. Financial viability of electric vehicle lithium-ion battery recycling. iScience 24(7), 102787. This is an average of LFP battery recycling costs based on China.* This value is not inclusive of the battery price, 125 \$/kWh, so it should be added when calculating the full life cycle of the battery. However, since 85 \$/kWh of the 125 \$/kWh is the cost of raw materials, this part of the cost will be replaced by recycled raw materials.

As well as the carbon emissions in the manufacturing and recycling stages of the battery life cycle, for new batteries, its value is 85 kgCO_{2e}/kWh from this reference: *Ciez, R.E., Whitacre, J.F., 2019. Examining different recycling processes for lithium-ion batteries. Nature Sustainability 2(2), 148-156.* 85 kgCO_{2e}/kWh represents the sum of carbon emissions generated by the LFP battery (the battery used in our paper) during the raw material mining (~51 kgCO_{2e}/kWh), manufacturing (~34 kgCO_{2e}/kWh) and

transportation ($<1 \text{ kgCO}_{2,e}/\text{kWh}$), which is **an average value for the United States**. This is due to the difficulty of obtaining data in China, the data of the United States is used for reference. The carbon emission during battery recycling, $69.8 \text{ kgCO}_{2,e}/\text{kWh}$ also comes from *Ciez, R.E., Whitacre, J.F., 2019. Examining different recycling processes for lithium-ion batteries. Nature Sustainability 2(2), 148-156.* As mentioned earlier, this data is an average value for the United States. Furthermore, this data ($69.8 \text{ kgCO}_{2,e}/\text{kWh}$) is also not included in the $85 \text{ kgCO}_{2,e}/\text{kWh}$. Considering that **$51 \text{ kgCO}_{2,e}/\text{kWh}$** are produced by raw materials mining, this part of carbon emissions will also be replaced by recycling carbon emissions.

If there is any more comment, please suggest them, and we will be very glad to make further improvements for this paper draft.

Additionally, there are sections of the report that are difficult to interpret, and this is likely due to the use of uncommon English phrases or vocabulary. As a result, reading the document to simply understand the intent of the statements took a substantial amount of time, without evaluating the technical quality of the work. Suggest that a native English speaker work with the authors to further refine the language for clarity. This will assist in future interpretation of the validity of the results.

Reply: In terms of language, we invited native English speakers to professionally revise our article, deleting or replacing many ambiguous words and phrases. All revisions are marked in the 'Manuscript_R1 with edition marks'.

Specific Comments:

- The V2B/B2V aspect of the paper is a particularly novel component, and yet there is no information provided on how this interaction works, what type of daily schedule it operates, how that affects depth of discharge of the batteries (which will affect degradation), etc. All that is stated is a general “multi-directional charging” statement, along with the results. The process and associated calculations must be explained in the paper, and referred to.

Reply: Thank you very much for your very professional comment. In order to explain or multi-directional V2X interaction, we have added a description in the Section-*Battery lifecycle and circular economy*:

‘The principle of multi-directional V2X interactions can be explained by Supplementary Note 2. When renewable energy is higher than the demand, the excess renewable energy is stored in battery energy storage systems (including electric vehicles). When renewable energy is lower than the demand, static batteries and electric vehicles will supply energy to cover the demand.’

Supplementary Figure 2 The operating principle of building-transportation energy sharing network.

Furthermore, due to the word limit of the main text, we describe it in more detail in the supplementary information, including the working principle of multi-directional V2X, vehicle operation schedules, and how the corresponding plan determines discharge depth and degradation degree of batteries:

‘The principles of V2B/B2V can be explained by Supplementary Fig. 2. In the building-transportation-storage energy system, renewable energy is divided into a surplus period and a shortage period, according to the relative difference between renewable energy and building energy demand. During the surplus period, excess renewable energy is stored in the battery energy storage system (including EVs) when the energy demand is completely covered. During the shortage period, building energy demands are firstly covered by renewable energy, and then by energy storage systems, before being covered by imported power from the electric grid. Excess renewable energy from the office building will be managed to charge EV group 1 or EV group 2 with the priority given to the EV groups with a lower FSOC. By contrast, the EV group with a higher FSOC will be given the priority for discharging to cover the demand shortage. An off-peak mandatory charging is applied to meet the vehicle’s travelling demands during the daytime. For EV group 1 and EV group 2, the maximum charge and discharge power are 10 kW and 20 kW, respectively. In addition, EVs only participate in V2B/B2V when parked next to the corresponding building.’ (SUPPLEMENTAL INFORMATION-Supplementary Note 2: The operating principle of building-transportation energy sharing network.)

Supplementary Figure 13 The travelling schedule of private cars. **a, b, c** The schedule of private cars (Group 1) on weekdays, Saturday and Sunday, respectively; **d, e, f** The schedule of shuttle buses (Group 2) on weekdays, Saturday and Sunday, respectively.

Supplementary Figure 3 The travelling schedule of private cars. **a, b, c** The Schedule 4+2 of private cars on weekdays, Saturday and Sunday, respectively; **d, e, f** The Schedule 8+6 of private cars on weekdays, Saturday and Sunday, respectively.

As can be seen from Supplementary Fig. 13, for private cars, their daily travelling time is 6 hours on weekdays, 4 hours on Saturdays, and does not travel on Sundays (Called “Schedule 6+4”). In order to

explore the impact of different scheduling on battery performance and carbon intensity, this study assumes two other schedules, as shown in Supplementary Fig. 3 (“Schedule 4+2” and “Schedule 8+6”). Then, they are compared with the case without multi-directional V2X, in terms of the depth of discharge and degradation degree. Results are shown in Supplementary Figs. 4 and 5.

As shown in Supplementary Fig. 4, when there is no multi-directional V2X, the depth of discharge is concentrated in the interval [0.1, 0.2) and [0.2, 0.3). After adding multi-directional V2X, regardless of the schedule of EVs, the distribution of its depth of discharge is more uniform compared to the case without multi-directional V2X, although the maximum distribution still occurs in [0.2, 0.3). In addition, scheduling has less impact on EV battery degradation. As shown in Supplementary Fig. 5, within 20 years of operation, except for the case without multi-directional V2X, all EV batteries need to be replaced three times.

Supplementary Figure 4 The depth of discharge distribution of different schedules. **a** No V2X; **b** Schedule 4+2; **c** Schedule 6+4; **d** Schedule 8+6.

Supplementary Figure 5 The evolution of degradation of different schedules. **a** No V2X; **b** Schedule 4+2; **c** Schedule 6+4; **d** Schedule 8+6.

(SUPPLEMENTAL INFORMATION-Supplementary Note 2: The operating principle of building-transportation energy sharing network.)

If there is any more comment, please suggest them, and we will be very glad to make further improvements for this paper draft.

• What is the charge/discharge cycle and depth of discharge for the EVs in the various scenarios? How might that affect the degradation, and then the carbon intensity?

Reply: Thank you very much for your very professional comment. Based on your comments, we've added the charge/discharge cycle and depth of discharge for the EVs in the various scenarios, and the impact on the degradation and the carbon intensity to the supplemental information:

In Supplementary Figs. 6 and 7, we summarize the charging and discharging cycles, degradation degree, and depth of discharge of EV batteries in different scenarios in Guangzhou. It is worth noting that in scenarios A, B, and D without multi-direction V2X, the number of battery cycles is much less than that in scenarios C and E with multi-direction V2X. In addition, the depth of discharge of scenarios A, B, and D is mainly concentrated at [0.1-0.3), which is the main depth of discharge when the vehicle is travelling, while the DOD distribution of scenarios C and E is more scattered because they begin to store excess renewable energy, and there is uncertainty in the amount of excess renewable energy. Since the degree of battery degradation is affected by a lot of factors, it is difficult to see from Supplementary Figs. 6 and 7 the specific effects of charge/discharge cycle and depth of discharge on battery degradation in different scenarios. Therefore, readers can refer to the battery degradation curve (Supplementary Fig. 8) to see the specific impact. From Supplementary Fig. 8, it's clear that a greater number of cycles will lead to higher battery degradation, and when the depth of discharge is far away from 0.4, the battery degradation will become even more obvious.

Supplementary Figure 6 The Equivalent number of cycles and evolution of relative capacity. a Equivalent number of cycles of private EVs in Guangzhou under different scenarios, **b** The evolution of relative capacity of private EV batteries in different scenarios.

(Note: Scenario A, B, D without multi-direction V2X interaction, Scenario C and E with V2X interaction. The equivalent number of cycles is reset around 1000 because a new battery is replaced.)

Supplementary Figure 7 DOD distribution of private electric vehicle batteries in different scenarios. a Scenario A, B, D; **b** Scenario C; **c** Scenario E.

Supplementary Figure 8 Battery degradation curves based on the depth of discharge and equivalent full cycles.

As for the battery carbon intensity, neither the charge/discharge cycles nor the depth of discharge has a direct relationship with the carbon intensity. This can be known from the results of Scenarios A, B, and D. Supplementary Table 1 summarizes the relevant results of battery carbon intensity in Guangzhou. It is noted that the carbon intensity varies greatly between different scenarios, with the minimum (Scenario B) at $-1,104.91 \text{ kg CO}_{2,e}/\text{kWh}$ and the maximum (Scenario A) at $1,625.01 \text{ kg CO}_{2,e}/\text{kWh}$. According to Supplementary Figs. 6 and 7, Scenario A and B have the same charge/discharge cycles and depth of discharge, but they have great differences in carbon intensity, which shows that the charge/discharge cycles and depth of discharge are not directly related to carbon intensity.

Supplementary Table 1 The carbon intensity of different scenarios in Guangzhou.

Scenarios	A	B	C	D	E
Carbon intensity ($\text{kg CO}_{2,e}/\text{kWh}$)	1625.01	-1104.91	108.04	504.06	-904.41

If there is any more comment, please suggest them, and we will be very glad to make further improvements for this paper draft.

- Top of page 8 statement that shuttle buses are “much lower” life cycle carbon emissions is overstated given the 10% difference

Reply: Thank you very much for your very professional comment. This is indeed an exaggeration, so we have changed it to “lower”. The revised expression is:

‘life-cycle of internal combustion engine buses (ICE-buses) at $721.3 \text{ g CO}_{2,e}/\text{km}$ is lower than those of electric buses (E-buses) at $715.7\text{-}784.4 \text{ g CO}_{2,e}/\text{km}$ ’

- Statements about the CER in the middle of page 8 discussion decreases in various regions, but not the time frame over which they are relevant or if they are part of the future hypothetical case study. If they are future values, are they aspirational or are there policy targets in place to achieve them

Reply: Thank you very much for your very professional comment. In Fig. 2b, the grey represents the CEF when there are only traditional coal power plants, while the green represents the CEF under the current energy structure (as shown in Fig. 2d). This is not a future-based value, but rather a calculated result based on the current local grid energy structure. In this Revision R1, we make it clear to help readers to easily follow.

Fig. 2 Comparison of lifecycle carbon emissions of electric vehicles (EVs) and internal combustion engine vehicles (ICEVs) under different grid energy compositions. a Carbon emission per kilometre of EVs and ICEVs in operation phase and life-cycle process under different climate zones; **b** Changes in

carbon emission factor from traditional grid to a clean grid; **c** Changes in the life-cycle carbon emissions of EVs and ICEVs with the transformation from a traditional grid to a clean grid; **d** Energy structures of the current clean grid in different climate zones.

If there is any more comment, please suggest them, and we will be very glad to make further improvements for this paper draft.

• Transportation of the renewable and battery systems was not included in the life cycle calculations. This is an important element to include. Perhaps it was not because the expectation is transport short distances within China due to active lithium ion battery industry there. Again, the choice to eliminate this life cycle phase must be addressed, and justified.

Reply: Thank you very much for your very professional comment. The transportation process is certainly important in lifecycle assessment. In this study, we calculated the carbon emissions of battery transportation by assuming that the average transportation distance of batteries in China is about 2000 km. The carbon emissions of battery transportation will only be less than 0.6 kg CO_{2,e}/kWh, which is far lower than the carbon emissions generated during its raw materials (51 kg CO_{2,e}/kWh), manufacturing (34 kg CO_{2,e}/kWh), operation and recycling (69.8 kg CO_{2,e}/kWh). Similar studies have also been done by other researchers. For example, this article^[A] proposes that carbon emissions in the transportation stage only account for 1% of the entire lifecycle of NMC batteries.

In addition, since Chinese battery companies tend to build factories in the local areas (such as BYD, CATL, Jinko, Gotion High-tech, etc.), their transportation distance will be far less than 2,000 km. Therefore, the carbon emissions during transportation are important with a small proportion. In this revision R1, we add the description as:

‘Furthermore, as carbon emissions in the transportation stage only account for 1% of the entire lifecycle of NMC batteries^[A] and uncertainties in specific transportation distances, transportation of the renewable and battery systems was not included in the life cycle calculations.’ (**BATTERY LIFE CYCLE AND CIRCULAR ECONOMY**)

Reference:

[A] Ciez RE, Whitacre JF. Examining different recycling processes for lithium-ion batteries. *Nature Sustainability* 2019; 2:148-56.)

If there is any more comment, please suggest them, and we will be very glad to make further improvements for this paper draft.

• In equations 2 the carbon emissions of recycling (or reuse) are included. It is unclear why, as this would normally be classified separately from materials processing and manufacture. Similarly, equation 12 in the appendix includes remanufacturing in the manufacturing calculation. There is an approach in the life cycle allocation of materials and outcomes during reuse of a product that assigns a fraction to the first life and a fraction to the second life. However, this is not explained at all in the paper and must be.

Reply: Thank you very much for your very professional comment. In this revision R1, we check carefully and further revise and explain the description, to make it more clear.

For equation 2, it is intended to represent the embodied carbon emissions we quote from PV and wind turbines arising from the manufacturing, operation and recycling stages. As shown in the Method section:

For PVs, the embodied carbon emission per kWp is shown in equation (10):^[A]

$$ECE_{PV, \text{unit}} = 560 \text{ kg CO}_{2,e}/\text{kWp} \quad (10)$$

For wind turbine, the embodied carbon per kW can be calculated by equation (11):

$$ECE_{WT, \text{unit}} = 1959.2 \times P_{WT, \text{rated}}^{-0.224} \quad (11)$$

where $ECE_{WT, \text{unit}}$ refers to the embodied carbon emission of the wind turbine with a unit of $\text{kg CO}_{2,e}/\text{kW}$, and $P_{WT, \text{rated}}$ refers to the rated power of the wind turbine.'

(Methods-IV) Embodied carbon of renewable energy)

Supplementary Figure 12 The embodied carbon emission of wind turbines. (SUPPLEMENTAL INFORMATION)

Equation (16) is obtained by fitting in Supplementary Fig. 12. The points in Supplementary Fig. 12 represent wind turbines with corresponding rated power and their embodied carbon emission with the data from Ref.^[B-H]

Since this article calculates the lifecycle carbon emissions of PV and wind turbines, based on empirical equations with embodied carbon emission per kWp, we have deleted equation 2 in the article.

For equation 12 (Now it's equation 7), the purpose is to calculate the carbon emissions of the battery in raw materials and manufacturing, remanufacturing phases. However, the original text was incorrectly stated, and CE_{manu} was repeatedly used with two different meanings. Therefore, we changed the original text of equation 12 to the following form:

To quantify the carbon emissions of batteries throughout the life cycle, the carbon intensity (CI) has been proposed as shown by equation (5):

$$CI = \frac{CE_{tot}}{Cap_{sys}} \quad (5)$$

where the total carbon emission (CE_{tot}) is as calculated by equation (6):

$$CE_{tot} = CE_{raw} + CE_{manu} + CE_{ope} + CE_{recyc} + CE_{RE} \quad (6)$$

where $CE_{raw-manu}$, CE_{ope} , CE_{recyc} and CE_{RE} refer to the carbon emission at the raw materials and manufacturing phase, operation phase, carbon emission of the recycling phase, embodied carbon emission of renewable energy systems. The specific calculations of $CE_{raw-manu}$, CE_{ope} , CE_{recyc} and CE_{RE} are as follows:

1) Raw materials and manufacturing phase

Carbon emission factor (CEF) is applied for carbon emission quantification with CEF_{raw} at 51 kg $CO_2, e/kWh$ from the raw materials, CEF_{manu} and CEF_{remanu} at 34 kg $CO_2, e/kWh$ from the manufacturing and remanufacturing process, respectively. Therefore, the carbon emission ($CE_{raw-manu}$) at the raw materials and manufacturing phase can be expressed by equation (7):

$$CE_{raw-manu} = CE_{raw} + CE_{manu} + CE_{remanu} \quad (7)$$

(Methodology-Battery carbon intensity calculation model)

Since equation 12 (now equation 7) only involves the raw material and manufacturing phases (including the remanufacturing phases of producing recycled materials into batteries), and does not involve stages such as recycling and reuse, this equation does not include $CE_{recycling}$ and CE_{ope} .

References:

- [A] Li G, Xuan Q, Pei G, Su Y, Lu Y, Ji J. Life-cycle assessment of a low-concentration PV module for building south wall integration in China. *Applied Energy*. 2018;215:174-85.
- [B] Vargas, A.V., Zenón, E., Oswald, U., Islas, J.M., Güereca, L.P., and Manzini, F.L. (2015). Life cycle assessment: A case study of two wind turbines used in Mexico. *Applied Thermal Engineering* 75, 1210-1216. 10.1016/j.applthermaleng.2014.10.056.
- [C] Smoucha, E.A., Fitzpatrick, K., Buckingham, S., and Knox, O.G. (2016). Life cycle analysis of the embodied carbon emissions from 14 wind turbines with rated powers between 50Kw and 3.4 Mw. *Journal of Fundamentals of Renewable Energy Applications* 6, 1000211.
- [D] Yang, J., and Chen, B. (2013). Integrated evaluation of embodied energy, greenhouse gas emission and economic performance of a typical wind farm in China. *Renewable and Sustainable Energy Reviews* 27, 559-568. 10.1016/j.rser.2013.07.024.
- [E] Chen, G.Q., Yang, Q., and Zhao, Y.H. (2011). Renewability of wind power in China: A case study of nonrenewable energy cost and greenhouse gas emission by a plant in Guangxi. *Renewable and Sustainable Energy Reviews* 15, 2322-2329. 10.1016/j.rser.2011.02.007.
- [F] Xie, J.-b., Fu, J.-x., Liu, S.-y., and Hwang, W.-s. (2020). Assessments of carbon footprint and energy analysis of three wind farms. *Journal of Cleaner Production* 254. 10.1016/j.jclepro.2020.120159.
- [G] Ji, S., and Chen, B. (2016). Carbon footprint accounting of a typical wind farm in China. *Applied Energy* 180, 416-423. 10.1016/j.apenergy.2016.07.114.
- [H] Savino, M.M., Manzini, R., Della Selva, V., and Accorsi, R. (2017). A new model for environmental and economic evaluation of renewable energy systems: The case of wind turbines. *Applied Energy* 189, 739-752. 10.1016/j.apenergy.2016.11.124.

If there is any more comment, please suggest them, and we will be very glad to make further improvements for this paper draft.

- Equation 2- what is the difference between manufacture and production?

Reply: Thank you very much for your very professional comment. After carefully checking the embodied carbon emission of renewable systems, we update the calculated method, which is calculated by empirical equations (Equations (10) and (11) in Methods) based on embodied carbon emission per kWp, involving manufacturing, transportation, operation, maintenance and recycling processes. As the meanings of manufacture and production are almost the same, we only keep the “manufacture/manufacturing” to avoid the repetition.

‘The embodied carbon of renewable energy equipment can be defined as the carbon emission generated during manufacturing, transportation, operation, maintenance and recycling of renewable energy equipment.

For PVs, the embodied carbon emission per kWp is shown in equation (10):³²

$$ECE_{PV, unit} = 560 \text{ kg CO}_{2,e}/\text{kWp} \quad (10)$$

For wind turbine, the embodied carbon per kW can be calculated by equation (11):

$$ECE_{WT, unit} = 1959.2 \times P_{WT, rated}^{-0.224} \quad (11)$$

where $ECE_{WT, unit}$ refers to the embodied carbon emission of the wind turbine with a unit of kg CO_{2,e}/kW, and $P_{WT, rated}$ refers to the rated power of the wind turbine.’ (*Section IV Embodied carbon of renewable energy-Methodology*)

In this revision R1, to make it more clear and avoid the misunderstanding, we revised as follows:

‘where ECE_{RE} is the embodied carbon emission of solar PV, BIPVs or wind turbines during their lifetime processes, which is calculated by empirical equations (equations (10) and (11) in Methods) based on embodied carbon emission per kWp, involving manufacturing, transportation, operation, maintenance and recycling processes.’ (*Embodied carbon emission factor (ECEf) on clean power production*)

Reference:

32 Li G, Xuan Q, Pei G, Su Y, Lu Y, Ji J. Life-cycle assessment of a low-concentration PV module for building south wall integration in China. *Applied Energy* 215, 174-185 (2018).

If there is any more comment, please suggest them, and we will be very glad to make further improvements for this paper draft.

- Page 11 states that zero-energy approach for the localized grid is chosen; that the total annual generation matches the annual electricity consumption. Is this realistic given today’s technologies or aspirational? If aspirational, it may account for what appear to be oversized renewables in Tables S22-24. For example, the US Department of Energy describes small scale wind turbines for residences in the range of 400W

to 20kW; the section in this paper is the maximum value. Similarly, hotels are expected to range from 1-100kW size, but this paper uses 500kW.

Reply: Thank you very much for your very professional comment. Zero-energy buildings and energy systems are achievable in reality, and have been piloted in different regions around the world (such as the CIC-ZCP in Hong Kong, and The Unisphere in the US). Their goal is for the building to generate energy equal to or greater than their own demands.

The reason why we adopt the wind turbine with large capacities is that, unlike the common low-rise residential buildings in the United States, more high-rise residential buildings (high energy demand) are located in China's cities, so larger-scale wind turbines are needed to maintain the zero-energy in buildings. By the way, it is possible because wind turbines can be installed not far from buildings, especially for cities close to sea or mountains (such as Shenzhen, Shanghai, Hong Kong, etc).

In addition, in terms of wind power products, companies worldwide have produced 20 kW or 500 kW, and associated technologies have become mature. The following table gives their parameter information and power-wind speed curve.

Parameters and wind speed-power curve of the 500-kW wind turbine

Parameters	Value
Wind turbine	Vestas V39
Rated power (kW)	500
Cut-in wind speed (m/s)	5
Rated speed (m/s)	15
Cut-out wind speed (m/s)	25
Hub height (m)	53
Rotor diameter (m)	39

Parameters and wind speed-power curve of the 20-kW wind turbine

Parameters	Value
Wind turbine	Hummer-h13.2-20kW
Rated power (kW)	20
Cut-in wind speed (m/s)	3
Rated speed (m/s)	9
Cut-out wind speed (m/s)	25
Hub height (m)	19.4
Rotor diameter (m)	13.2

If there is any more comment, please suggest them, and we will be very glad to make further improvements for this paper draft.

• Page 11 – why is scenario 2 pure renewables a solar scenario, when it is clear from the other data and figures in the paper that wind is a much more common and valued resource in China?

Reply: Thank you very much for your very professional comment. Solar energy is chosen for the analysis of Scenario B because that, compared to wind energy, the solar energy is more temporally dependent, i.e., clean power supply at daytime when the sun rises, but no clean energy at nighttime during the sunset. Solar energy can be a more suitable candidate for spatiotemporal energy analysis with energy storages. Furthermore, although wind power resources are abundant in China, its resources are mainly abundant

in sparsely populated areas such as the west and north of China, while the wind resources are not abundant or not easily utilised in the densely populated southeastern areas, due to the blocking from high-rise buildings. Contrarily, solar energy resources can be easily utilized in both distributed and centralised forms, thus realizing ideal scenarios.

If there is any more comment, please suggest them, and we will be very glad to make further improvements for this paper draft.

• Page 13 – the paper notes “electricity prices are generally higher than the costs of solar power”. This is an unusual statement from the point of view of certain countries and grid design. Can the reasoning behind this be explained?

Reply: Thank you very much for your very professional comment. In fact, this sentence analyses different regions in China. Since the price of PV panels sold in China is very cheap, the LCOE of photovoltaic panels is quite low. We have made a comparison of the local average electricity price and the LCOE of PV power generation, as shown in the figure:

It is noted that in the seven representative different regions in China that we calculated, the LCOE of PV power generation is lower than the average electricity price. Therefore, we describe in this article as ‘electricity prices are generally higher than the costs of solar power’.

If there is any more comment, please suggest them, and we will be very glad to make further improvements for this paper draft.

• Page 13-14 – sentence does not make sense “...the overwhelmed dominance of the import cost savings over the battery cycling ageing costs”. Does this intend to state the degradation is slower in this scenario? What is an “import cost savings”?

Reply: Thank you very much for your very professional comment. The “import cost saving” shows the reduced amount of electricity imported from the power grid multiplied by the electricity price. Specifically, energy storage batteries and EV batteries store part of the renewable energy during the renewable surplus period and release the energy during the shortage period, reducing the electricity amount imported from the power grid. Battery cycling ageing cost refers to the replacement, recycling, remanufacturing and repurposing cost of the battery after its ageing.

Specifically, the table below shows the import cost savings and battery cycling ageing in different cities in Scenario C. It can be seen that in most cities, the import cost saving will be higher than the battery cycling ageing cost:

If there is any more comment, please suggest them, and we will be very glad to make further improvements for this paper draft.

• Page 14- why is the cost of the reused battery higher?

Reply: Thank you very much for your very professional comment. In this study, the cost of the reused battery is much lower than the new batteries. In fact, the cost of reused batteries is lower. The literature we cited shows that the cost of reused batteries is 35 US\$/kWh, which is much lower than the 125 US\$/kWh of new batteries. In this Revision R1, we add the information to avoid the misunderstanding.

The price of reused batteries is from: *Neubauer J, Smith K, Wood E, Pesaran A. Identifying and Overcoming Critical Barriers to Widespread Second Use of PEV Batteries. United States 2015. p. Medium: ED; Size: 93 p.*

The price of new batteries is from: *Deshwal D, Sangwan P, Dahiya N. Economic Analysis of Lithium Ion Battery Recycling in India. Wireless Personal Communications. 2022;124:3263-86.*

If there is any more comment, please suggest them, and we will be very glad to make further improvements for this paper draft.

• Some graphics have acronyms that are not defined (e.g., ST, RC). Please define them.

Reply: Thank you very much for your very professional comment. We have modified the expressions of ST and RC. In addition, ST means “solar thermal collector”, while RC means “relative capacity”. All revisions are marked in the ‘Manuscript_R1 with edition marks’.

If there is any more comment, please suggest them, and we will be very glad to make further improvements for this paper draft.

• Figure 6 – I am confused how the life cycle carbon intensity goes negative in panel B. Should it not hold constant rather than increase?

Reply: Thank you very much for your very professional comment. Here, lifecycle carbon intensity refers to the battery lifecycle, including raw material collection, manufacturing, operation, recycling and remanufacturing phases. The raw material collection, manufacturing, recycling and remanufacturing stages of batteries are all carbon positive. However, in Scenario B, since the battery will store renewable electricity for use by the EV during the operation phase, this means that less traditional energy power is used, thereby indirectly reducing carbon emissions. Therefore, the carbon emissions during the battery operation phase can be expressed by the following equation:

$$CE_{ope} = \sum_{i=1}^j \int_0^{t_{end}} [P_{grid,ch,i}(t) - P_{RE,ch,i}(t)] \cdot CEF_{coal} dt \quad (8)$$

Where $P_{grid,ch,i}(t)$, $P_{RE,ch,i}(t)$, and CEF_{grid} refer to the grid power charged to the battery, the renewable energy charged to the battery, and the carbon emission factor of the coal-power based grid.

Since in scenario B, the battery is only charged with renewable energy, and the carbon emissions during the operation phase is negative, resulting in the final total carbon intensity to be negative.

This concept is consistent with the ‘IEA EBC Annex 87 Positive Energy Districts [B]’, in which the carbon emission is calculated by:

$$CE_{community} = \int_0^{t_{end}} [P_{imp}(t) - P_{exp}(t)] \cdot CEF dt$$

In the Positive Energy Districts, as the $P_{exp}(t)$ is much higher than the $P_{imp}(t)$, carbon emission of the community ($CE_{community}$) is negative.

Reference:

[B] IEA EBC Annex 83: Positive Energy Districts. <https://annex83.iea-ebc.org/>

If there is any more comment, please suggest them, and we will be very glad to make further improvements for this paper draft.

• Page 22 – the battery degradation data used was based on a first life situation, which is fine. However, the paper does not state what degradation data was used for the second life / reuse situation.

Reply: Thank you very much for your very professional comment. The battery degradation data we use comes from ^[1], which is explained in our original article as follows:

The battery model used in this paper is extended from a previously developed battery model.²⁶ Supplementary Figure 9 shows the degradation curve of the battery model, which is characterized by the different degradation speeds of the battery under different depth of discharge (DoD). According to this characteristic, a system based on number of cycles and dynamic DoD is developed. The dynamic DoD in Li-ion batteries (LFP batteries) can be calculated by Equation (3):

$$DoD = FSOC_{peak,cycle n} - FSOC_{valley,cycle n} \quad (3)$$

where $FSOC_{peak,cycle n}$ and $FSOC_{valley,cycle n}$ refer to the highest and the lowest point of FSOC at the n^{th} cycle.

Supplementary Figure 9 Battery degradation curves based on the depth of discharge and number of cycles.

In addition, according to Supplementary Figure 9, relative capacity (RC) is used to illustrate the degree of degradation of lithium-ion batteries. The Li-ion battery degradation curve can be polynomial fitted with exponential power at 3 according to the data in Ref.^[1], which is expressed in Equation (4).

$$RC_{DoD} = k_1 CycleNum^3 + k_2 CycleNum^2 + k_3 CycleNum + k_4 \quad (4)$$

Table 1 Polynomial coefficients of the battery degradation model.

	DoD = 1	DoD = 0.8	DoD = 0.6	DoD = 0.4	DoD = 0.3	DoD = 0.2	DoD = 0.1
k_1	-2.685×10^{-11}	-8.732×10^{-12}	-2.562×10^{-12}	-5.362×10^{-13}	-3.084×10^{-13}	-1.934×10^{-13}	-1.292×10^{-13}
k_2	1.539×10^{-7}	6.271×10^{-8}	2.665×10^{-8}	9.537×10^{-9}	6.622×10^{-9}	4.866×10^{-9}	3.727×10^{-9}
k_3	-3.261×10^{-4}	-1.947×10^{-4}	-1.276×10^{-4}	-7.764×10^{-5}	-6.492×10^{-5}	-5.581×10^{-5}	-4.894×10^{-5}
k_4	1	1	1	1	1	1	1

This degradation data includes the SOH from 100% to 60%. We have referred to Ref. [1] and set the SOH of first life as 100%-80% and the second life as 80%-60%. The above battery degradation data is used for the second life.

References:

[I] Yan G, Liu D, Li J, Mu G. A cost accounting method of the Li-ion battery energy storage system for frequency regulation considering the effect of life degradation. Protection and Control of Modern Power Systems. 2018;3.
 [J] Shafique M, Rafiq M, Azam A, Luo X. Material flow analysis for end-of-life lithium-ion batteries from battery electric vehicles in the USA and China. Resources, Conservation and Recycling. 2022;178.

If there is any more comment, please suggest them, and we will be very glad to make further improvements for this paper draft.

REVIEWERS' COMMENTS

Reviewer #2 (Remarks to the Author):

After careful review of the revised manuscript draft, the reviewer wants to thank the authors for satisfactorily addressing each and every review comment of the reviewer. And the reviewer wishes all the best to the authors for all their future endeavors.

Reviewer #3 (Remarks to the Author):

Thank you for the very detailed and thorough consideration of my previous review. My comments and concerns have been addressed, and I believe the manuscript is ready for publication.